# Characterizing sleep spindles in 11,630 individuals from the National Sleep Research Resource

S.M. Purcell[1,2,3], D.S. Manoach[2,4,5], C. Demanuele[2,4,5], B.E. Cade[6,7], S. Mariani[6,7], R. Cox[2,8], G. Panagiotaropoulou[2,4,5], R. Saxena[9,10,11], J.Q. Pan[12], J.W. Smoller[2,4,13], S. Redline[2,6,7,*] & R. Stickgold[2,8,*]

Sleep spindles are characteristic electroencephalogram (EEG) signatures of stage 2 non-rapid eye movement sleep. Implicated in sleep regulation and cognitive functioning, spindles may represent heritable biomarkers of neuropsychiatric disease. Here we characterize spindles in 11,630 individuals aged 4 to 97 years, as a prelude to future genetic studies. Spindle properties are highly reliable but exhibit distinct developmental trajectories. Across the night, we observe complex patterns of age- and frequency-dependent dynamics, including signatures of circadian modulation. We identify previously unappreciated correlates of spindle activity, including confounding by body mass index mediated by cardiac interference in the EEG. After taking account of these confounds, genetic factors significantly contribute to spindle and spectral sleep traits. Finally, we consider topographical differences and critical measurement issues. Taken together, our findings will lead to an increased understanding of the genetic architecture of sleep spindles and their relation to behavioural and health outcomes, including neuropsychiatric disorders.

[1] Department of Psychiatry, Brigham and Women's Hospital, Boston, Massachusetts 02115, USA. [2] Harvard Medical School, Boston, Massachusetts 02115, USA. [3] Department of Psychiatry, Icahn School of Medicine at Mount Sinai, New York, New York 10029, USA. [4] Department of Psychiatry, Massachusetts General Hospital, Boston, Massachusetts 02114, USA. [5] Athinoula A. Martinos Center for Biomedical Imaging, Charlestown, Massachusetts 02129, USA. [6] Division of Sleep and Circadian Disorders, Brigham and Women's Hospital, Boston, Massachusetts 02115, USA. [7] Division of Sleep Medicine, Harvard Medical School, Boston, Massachusetts 02115, USA. [8] Department of Psychiatry, Beth Israel Deaconess Medical Center, Boston, Massachusetts 02215, USA. [9] Center for Human Genetic Research, Massachusetts General Hospital, Boston, Massachusetts 02114, USA. [10] Department of Anesthesia, Critical Care and Pain Medicine, Massachusetts General Hospital, Boston, Massachusetts 02114, USA. [11] Program in Medical and Population Genetics, Broad Institute, Cambridge, Massachusetts 02142, USA. [12] Stanley Center for Psychiatric Research, Broad Institute of Harvard and MIT, Cambridge, Massachusetts 02142, USA. [13] Psychiatric and Neurodevelopmental Genetics Unit, Massachusetts General Hospital, Boston, Massachusetts 02114, USA. * These authors contributed equally to this work. Correspondence and requests for materials should be addressed to S.M.P. (email: smpurcell@bwh.harvard.edu).

Sleep spindles—bursts of 11–15 Hz (sigma frequency band) activity, typically between 0.5 and 2 s in duration—are characteristic transient features of the sleep electroencephalogram (EEG). Spindles are most prominent during N2 sleep and are in fact a defining feature of this stage. Although their underlying neural circuitry has been relatively well characterized—being generated in the thalamic reticular nucleus and synchronized by thalamocortical interactions[1]—their function is less clearly defined. Multiple lines of observational[2] and experimental[3,4] evidence point to an important role of sleep spindles in normal memory and learning[5], and they are believed to be influenced by, or to partly mediate, dynamic alterations in synaptic plasticity seen during sleep[6,7].

Within individuals, spectral properties of the sleep EEG including sigma power are highly stable (fingerprint-like) across nights[8–10], even under markedly different sleeping conditions[11]. Between individuals, there is considerable variability in both the typical quantity and quality of spindles, in part attributable to demographic factors such as age[12,13] or sex[14,15]. Genetic factors also play a role, as evidenced by twin studies of EEG power[16,17] and spindles[18]. However, previous studies have been limited by sample sizes typically two or three orders of magnitude smaller than the one described here.

Genetic studies are further motivated by suggestions that spindle deficits are biomarkers for neuropsychiatric diseases, including schizophrenia[19–23]. Spindle activity is modifiable[24–27] and could represent an attractive therapeutic target if causally implicated in disease risk. In light of recent progress in schizophrenia genetics[28,29], genetically characterizing variation in spindle activity could help to elucidate pathways between genetic risk and neuropsychiatric disease[23]. If spindles, or other aspects of sleep, are causally related to disease, it could have both diagnostic and therapeutic implications, as others have suggested for Alzheimer's disease, in which predictive sleep characteristics may precede cognitive symptoms[30].

Well-powered genetic studies of sleep are emerging, in particular for self-reported traits including chronotype (morning versus evening preference) and sleep duration[31–33]. As for any complex trait, large sample sizes will be necessary to detect genes of individually modest effects[34]. Genetic studies of spindles will require fully automated processing and the use of large convenience samples with potentially noisy data sources. With this in mind, here we detect and characterize spindles in a large sample, which allows us to identify factors associated with their variation, to corroborate some previously reported phenomena and to establish robust, heritable measures for subsequent molecular genetic studies.

An immediate challenge is that spindle activity cannot unambiguously be reduced to a single quantitative trait, as it likely represents a complex and possibly genetically heterogeneous set of processes. What are the optimal and independent aspects of spindle activity: density, duration, amplitude, frequency or other features? Does it matter when spindles occur: earlier versus later in the night, or in the life course? Where should spindle activity be measured, given clear topographical differences? Do inferences in healthy individuals generalize to patients, or are there other latent group differences that may impact genetic studies?

Here we investigate some of these questions as a prelude to future molecular genetic studies, in an analysis of over 10,000 individuals to characterize normative distributions and epidemiological associations of spindle activity. In particular, we demonstrate the considerable heterogeneity that exists—both between and within individuals—as well as establishing a heritable basis for individual differences.

## Results

**Overview of studies.** We combined polysomnography and demographic, anthropometric and medical history data on 11,630 individuals aged 4 to 97 years (Table 1 and Supplementary Fig. 1) from the National Sleep Research Resource (NSRR)[35,36]. After filtering to include only epochs of N2 sleep without manually annotated arousals, movements or artefacts, performing a series of statistical filters (Supplementary Table 1 and Supplementary Figs 2–7) and correcting for cardiac interference, the final spindle data set was created based on a total of 16,499 h of N2 sleep, or 1.4 h per individual (see the Methods section for details).

**Analysis of canonical spindles.** In an initial analysis (denoted below as the canonical analysis), we broadly targeted spindle activity in the sigma range, centred on 13.5 Hz, and detected a total of 3,846,408 spindles across C3 and C4 EEG channels. The mean spindle density, averaged over both channels, was 1.88 spindles per minute of artefact-free N2 sleep, comparable to other reports in healthy adults[37–39]. Figure 1a shows the distribution of spindle properties including density (count per minute), amplitude, duration and frequency (see also Supplementary Table 2). These properties were highly intercorrelated (Supplementary Table 3), which in part reflects the differential detectability of spindles as a function of their true amplitude, duration or frequency, as discussed below. We observed a high positive correlation between spindle density and sigma power (Pearson's $r = 0.52$, $P < 10^{-15}$, $N = 11,148$; Supplementary Fig. 8), especially between an individual's mean spindle amplitude and sigma power (Pearson's $r = 0.95$, $P < 10^{-15}$, $N = 11,146$).

To assess stability over time, we calculated test–retest correlations in 4,079 individuals for whom a second night of polysomnography was available. Children in the Childhood Adenotonsillectomy Trial (CHAT) study were retested ∼6 months after their initial polysomnogram; adults in Sleep Heart Health Study (SHHS) and Osteoporotic Fractures in Men ancillary sleep study (MrOS) were retested after ∼6 years. For spindle density, test–retest correlations were $r = 0.74$ ($N = 245$), 0.81 ($N = 2,597$) and 0.76 ($N = 958$) for CHAT, SHHS and MrOS, respectively (Pearson's correlations, all $P < 10^{-15}$; see Fig. 1b) and remained high after adjustment for age, sex and race. Other spindle properties showed substantial test–retest correlations also (Supplementary Table 4 and Supplementary Fig. 9), suggesting that our measures are stable traits, reliably measured and amenable to genetic analyses.

**Sex differences.** Females had 0.16 more spindles per minute than males ($P < 10^{-15}$, $N = 10,387$, linear regression of spindle density on sex along with other covariates, see Methods section). Although menstrual cycles may impact spindle activity[40], information on menstrual timing was not available (although note that the majority of females were outside the reproductive age range). Between ages 15 and 45 years, we observed a significantly greater variance (F-test to compare variances, $P = 0.016$) in spindle density estimates for females ($s^2 = 0.69$, $N = 650$) than males ($s^2 = 0.57$, $N = 587$), consistent with a sex-specific effect on spindle activity that introduces more variability in females.

**Sleep macroarchitecture and spindles.** Older individuals and males had less deep, N3 sleep (Supplementary Table 5). However, males tended to have more N2 sleep (as measured by both the absolute number of minutes and the proportion of total sleep time). Older individuals had less sleep in total and that was also less deep, with fewer minutes of N2, but even greater per cent reductions in N3 and rapid eye movement (REM). Animal studies

**Table 1 | Demographic and sleep summaries for National Sleep Research Resource studies.**

| | CHAT (childhood) | CCSHS (adolescence) | CFS (lifespan) | SHHS (middle-age) | MrOS (late adulthood) | SOF (late adulthood) | Combined (lifespan) |
|---|---|---|---|---|---|---|---|
| *Sample size* | | | | | | | |
| First visit | 1,232 | 515 | 730 | 5,793 | 2,907 | 453 | 11,630 |
| Second visit | 407 | 0 | 0 | 2,647 | 1,025 | 0 | 4,079 |
| *Sex* | | | | | | | |
| Female | 52% | 50% | 55% | 52% | 0% | 100% | 41% |
| *Age (years)* | | | | | | | |
| Mean | 7.0 | 17.7 | 41.4 | 64.5 | 77.6 | 82.9 | 58.0 |
| *Race* | | | | | | | |
| White | 39% | 60% | 41% | 85% | 91% | 92% | 9,055 |
| Black | 47% | 36% | 56% | 9% | 3% | 8% | 1,815 |
| Other | 14% | 4% | 3% | 7% | 6% | 0% | 760 |
| *Sleep architecture* | | | | | | | |
| Total sleep time (h) | 7.53 | 7.76 | 6.22 | 6.00 | 6.02 | 5.79 | 6.25 |
| N1 sleep | 8.3% | 4.2% | 5.2% | 5.3% | 6.8% | 5.3% | 5.9% |
| N2 sleep | 41.8% | 52.0% | 56.7% | 57.6% | 62.0% | 55.9% | 56.7% |
| N3 sleep | 31.6% | 23.2% | 20.2% | 17.7% | 11.1% | 20.6% | 18.0% |
| REM sleep | 18.3% | 20.6% | 17.9% | 19.3% | 18.8% | 18.2% | 19.0% |
| N2 sleep pre-QC (h) | 3.15 | 4.04 | 3.50 | 3.45 | 3.72 | 3.24 | 3.51 |
| N2 sleep post-QC (h) | 1.65 | 1.97 | 1.50 | 1.53 | 1.21 | 1.07 | 1.46 |

CCSHS, Cleveland Children's Sleep and Health Study; CFS, Cleveland Family Study; CHAT, Childhood Adenotonsillectomy Trial; MrOS, Osteoporotic Fractures in Men ancillary sleep study; QC, quality control; SHHS, Sleep Heart Health Study; SOF, Study of Osteoporotic Fractures.
For 11,630 independent individuals (4,079 of whom had two nights of polysomnography, typically after more than a 5-year interval), summary statistics are tabulated from the first, baseline visit. The six studies span a considerable range of ages and most studies have substantial within-study variability in age, with the exception of CCSHS. Four studies have approximately equal numbers of males and females, except MrOS (males only) and SOF (females only). Race is based on self-report and is summarized as 'white' (American of European self-reported ancestry/ethnicity), 'black' (typically American of African self-reported ancestry/ethnicity) and 'other' (typically American of self-reported Asian ancestry or Hispanic/Latino ethnicity). Sleep was staged manually by expert polysomnographists. The average number of hours of N2 sleep is presented before any filtering/QC, as well as after QC; the final row indicates the average duration of N2 sleep used in subsequent spectral and spindle analyses (1.4 h per individual on average).

have suggested that spindles play a direct role in regulating sleep architecture, as optogenetically induced spindles increased the duration of non-rapid eye movement (NREM) sleep in mice[41]. Consistent with this, we observed that higher spindle density during N2 was associated with a greater duration of N2 sleep ($P = 1 \times 10^{-11}$, 4.4 min per unit increase in spindle density, $N = 10,390$), in a linear regression with N2 duration as the dependent variable, controlling for age, sex, race, arousal index, apnoea–hypopnoea index (AHI) and study membership (Supplementary Table 6). Applying a similar analysis, higher spindle density in N2 was also associated with a shorter duration of N3 sleep ($P < 10^{-15}$, 5.2 min decrease per unit spindle density, $N = 10,348$) and a greater duration of REM sleep ($P = 3 \times 10^{-6}$, 1.5 min increase per unit spindle density, $N = 10,399$). The latter effects were stronger in older individuals, with interactions in predicting N3 ($P = 1 \times 10^{-6}$) and REM ($P = 2 \times 10^{-10}$) sleep obtained by adding an age-by-spindle density term to each of the above regression models. (Supplementary Table 7).

**Life-course trajectories of spindle traits.** Spindle activity changes with age, which is believed to reflect the maturation and later development or disruption of thalamocortical regulatory mechanisms[42]. That spindles change with ageing is not inconsistent with the presence of substantial and stable individual differences: height, for example, is a highly heritable trait that changes both dramatically and predictably with age. In adults, most studies report a progressive decline in spindle density or sigma power with increasing age[39,43–47] and an increase in spindle oscillatory frequency[44–46]. In contrast, most studies in children and adolescents report an increase in sigma power or spindle activity with increasing age[12,48]. Other studies

have reported an increase in peak sigma frequency from early childhood into adolescence[49,50].

We assessed age-related changes in three ways, each yielding similar results: in a cross-sectional analysis across the entire sample (Supplementary Fig. 10), in cross-sectional analyses within each study (Supplementary Table 8) and longitudinally (Supplementary Table 9). Spindle density increased with age during childhood, peaked around adolescence and then declined across adulthood (Supplementary Fig. 10). In contrast, spindle duration progressively declined from young childhood to late adulthood. Spindle amplitude was broadly stable across childhood and adolescence, but declined in adulthood. Spindle frequency increased during childhood and plateaued in adulthood.

Within-study statistical modelling produced consistent results: spindle density significantly increased with age during childhood (CHAT study) but declined in all the adult studies (Supplementary Table 8). Other sleep parameters were correlated with age, including sleep efficiency. Although sleep efficiency was positively correlated with spindle density (Pearson's $r = 0.06$, $P = 9 \times 10^{-10}$, $N = 8,216$), there was, as others have found[39], no independent effect of sleep efficiency on spindles after controlling for age.

As previous reports have suggested that the decline in spindle density from middle to late adulthood is attenuated in females[46], we formally tested for age-by-sex interaction within the two adult samples containing both males and females (Cleveland Family Study (CFS) and SHHS). We observed nominally significant interaction effects in both studies, consistent with relatively less decline in females (72% and 90% of the decline in males for CFS and SHHS, $N = 719$ and 5,572, respectively, both $P < 0.05$ from a linear regression on spindle density, controlling for standard covariates).

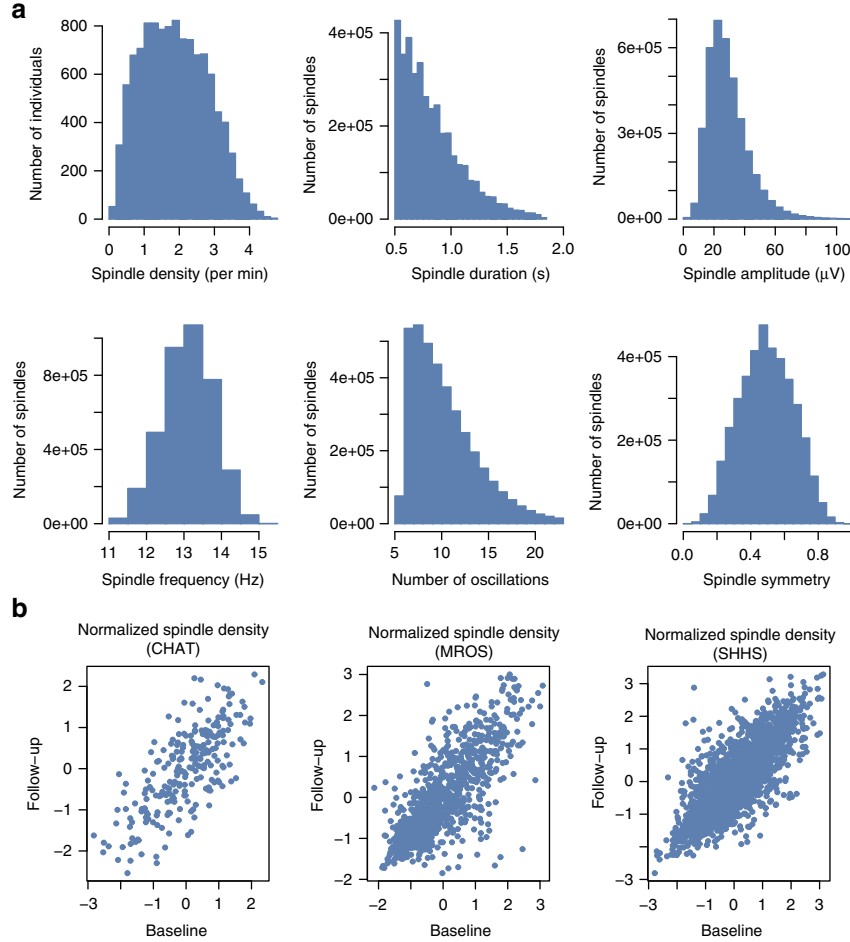

**Figure 1 | Spindle trait distributions at baseline and across time.** (**a**) Distribution of spindle density (spindles per min), duration (s), amplitude (μV), frequency (Hz), number of oscillations and symmetry index for all 11,630 individuals at baseline. (**b**) Scatter plots of baseline versus follow-up estimates of spindle density for individuals in the three studies (CHAT, SHHS and MrOS) with repeated polysomnography, showing standardized residuals from a linear regression of spindle density on age, sex and race by linear regression. Spindle density and other spindle traits exhibited high test–retest correlations, with or without adjustment for age, sex and race (Supplementary Table 4 and Supplementary Fig. 9).

**Longitudinal analyses.** We observed similar patterns of results in the longitudinal, within-individual analyses (Supplementary Table 9 and Fig. 1b), which are statistically independent of the cross-sectional analyses as no second-visit polysomnograms were included in the primary analyses. Despite a relatively small sample size and brief test–retest interval (∼6 months), we observed a nominally significant 3% increase in spindle density between the two polysomnograms in the childhood CHAT study (matched-pair $t$-test $P = 0.023$, $N = 245$). In contrast, both adult studies showed highly significant (all $P < 10^{-15}$, matched-pair $t$-tests) reductions in spindle density (12% and 15% reductions in SHHS and MrOS, $N = 2,597$ and 958, respectively), amplitude (5 and 12%) and duration (2 and 2%) across the ∼5-year period, consistent with the cross-sectional analyses.

**Frequency-dependent spindle analyses.** Up to this point, we have only considered spindles from the canonical wavelet analyses that targeted activity broadly centred around 13.5 Hz. There is neither consensus nor objective data on what constitutes the true range of frequencies for spindles; however, human studies have used different criteria, with lower bounds as low as 9 Hz[37–51] and upper bounds as high as 18 Hz[52]; animal studies report spindle frequencies as low as 6 or 7 Hz[53,54]. Furthermore, multiple studies have argued for two types of sleep spindles, with

qualitatively distinct topographical and functional associations: fast spindles (above ∼13 Hz) occurring primarily over centroparietal derivations, and slow spindles (below ∼13 Hz) occurring more often at frontal derivations[10,55–57]. Functional magnetic resonance imaging also suggests that fast and slow spindles have different cortical sources[58].

To target a greater range of spindle frequencies, we set the wavelet's $F_C = 8$ to 18 Hz in 0.25 Hz increments, yielding a vector of spindle density estimates per individual in a frequency-dependent analysis. Across this entire range of frequencies, the extent to which the transient features detected in the EEG as spindles in fact represent the same underlying processes is an open question. Similarly, signal-to-noise ratios may vary when targeting spindles across this broad range of frequencies. To address these problems empirically, we therefore explicitly detected spindles at different frequencies and analysed them separately.

Spindles from the frequency-dependent analysis appeared to be reliably measured and relatively stable over time (Supplementary Figs 11–13). Figure 2 recasts the analysis of life-course trajectories in terms of vectors of frequency-dependent spindle densities. The predominant trend was for spindles to increase in frequency during childhood until early adulthood and thereafter decline in density, with the modal spindle frequency remaining stable. Slower spindles (for example, $F_C = 11$ Hz) peak earlier in

development than faster spindles (for example, $F_C = 15$ Hz), consistent with earlier studies documenting an upward shift in the sigma peak during childhood and adolescence[12,50]. These results suggest that it will be challenging to interpret variation in spindle density if age and spindle frequency are not taken into account. Slower spindles also did not exhibit sex differences in density ($P = 0.63$, $N = 10,182$, linear regression of spindle density on standard covariates; Supplementary Table 10), whereas faster spindles were denser in females (0.3 spindles per minute more, $P < 10^{-15}$, $N = 10,179$), consistently observed across studies (Supplementary Fig. 12).

Figure 2a also indicates activity in the alpha-range (8–9 Hz), which appears as a separate cluster and is largely restricted to young adulthood; the exact relationship of this slower spindle-like activity to more commonly defined slow (for example, 11 Hz) and fast (for example, 15 Hz) spindles is unclear. Below, when referring to slow spindles generically, we imply spindles around 11 Hz, following most previous studies.

**Spindles during slow-wave sleep.** Spindles occurred less often in N3 than in N2 sleep (1.45 versus 1.88 spindles per minute, paired $t$-test $P < 10^{-15}$, $N = 7,716$), especially so in younger individuals (Supplementary Table 11). N2 and N3 spindle density estimates were nonetheless strongly correlated (Pearson's correlation $r = 0.77$, $P < 10^{-15}$, $N = 7,716$). Spindles during N2 and N3 sleep generally showed similar demographic associations (Supplementary Table 11), with the qualification that the sex

effect was stronger, whereas the association with age was weaker. However, whereas spindle density during N2 significantly predicted the duration of N2, N3 and REM sleep, spindle density during N3 was unrelated to sleep macroarchitecture (Supplementary Table 6). In a subset of SHHS individuals for whom staging distinguished NREM 3 and NREM 4 sleep, we observed qualitatively similar results for spindles detected during NREM 4 sleep (Supplementary Fig. 14), underscoring that spindle activity is not specific to N2 sleep.

**Changes in spindle activity between NREM sleep cycles.** With regard to the dynamics of spindles across the night, some studies have reported a decrease in spindle density[56], whereas others have reported an increase. How spindle activity changes over the night has been reported to be dependent on age[43,47,59], with temporal effects being attenuated with increasing age. Refocusing only on N2 sleep, we estimated spindle density separately for each NREM sleep cycle[60], which pointed to qualitatively different, age-dependent dynamics for fast and slow spindles (Fig. 3a). Spindle activity during the first cycle seemed exceptional, in that there were simple linear trends across cycles 2–5, whereby younger individuals showed greater increases in fast spindle activity across the night, but older individuals showed greater decreases in slow spindle activity across the night. At least in younger individuals, the greater increase in fast compared to slow spindles is consistent with previous reports of increases in spindle frequency over the course of the night[10,61].

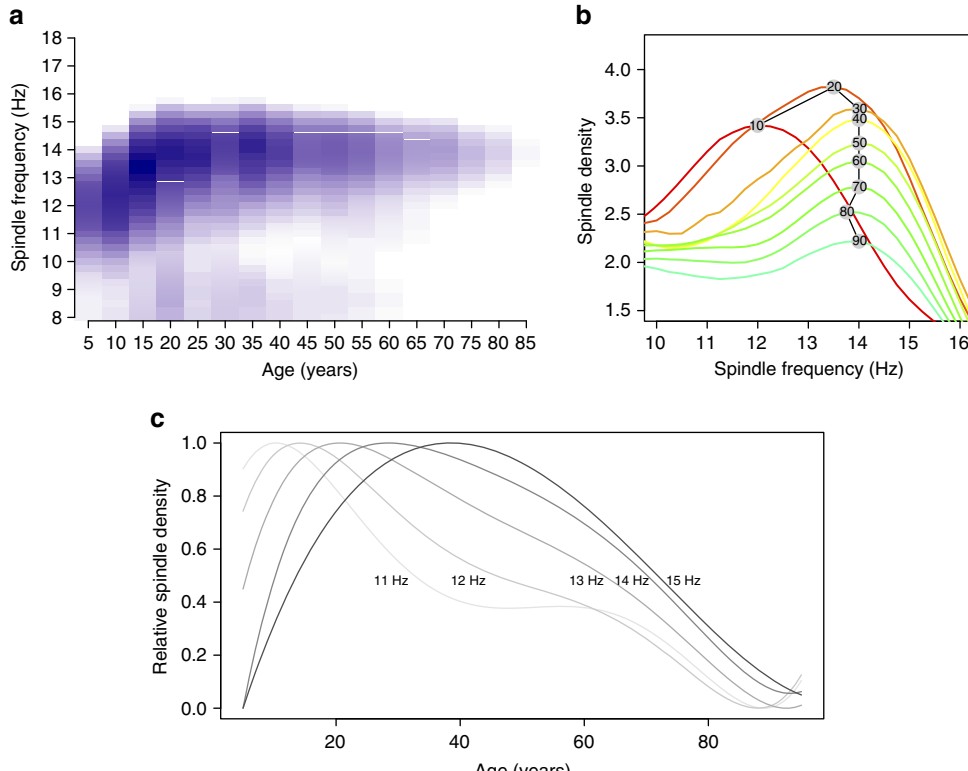

**Figure 2 | Changes in spindle frequency and density with age.** These plots depict average spindle density estimates (spindles per min) from the variable-frequency analysis (setting the wavelet $F_C = 8$ to 18 Hz in 0.25 Hz intervals). (**a**) Darker shades indicate greater spindle density. The predominant trend is for spindles to increase in frequency during childhood until early adulthood and thereafter decline in density (with the modal spindle frequency remaining stable). (**b**) Based on the same data as (**a**), each line represents the mean spindle density (y-axis) for individuals grouped by age range (10 = 5–15 years; 20 = 16–25 years, and so on), plotted as a function of targeted spindle frequency ($F_C$) on the x-axis, reinforcing the qualitatively different changes in spindle frequency and density during childhood versus adulthood. (**c**) Age-dependent expectations for spindle densities for a core range of targeted frequencies (11–15 Hz), derived from a linear model describing spindle density as a nonlinear function of age and other covariates, with each curve normalized to have a maximum of 1.0. Slower spindles (for example, $F_C = 11$ Hz) peak earlier in development than faster spindles (for example, $F_C = 15$ Hz).

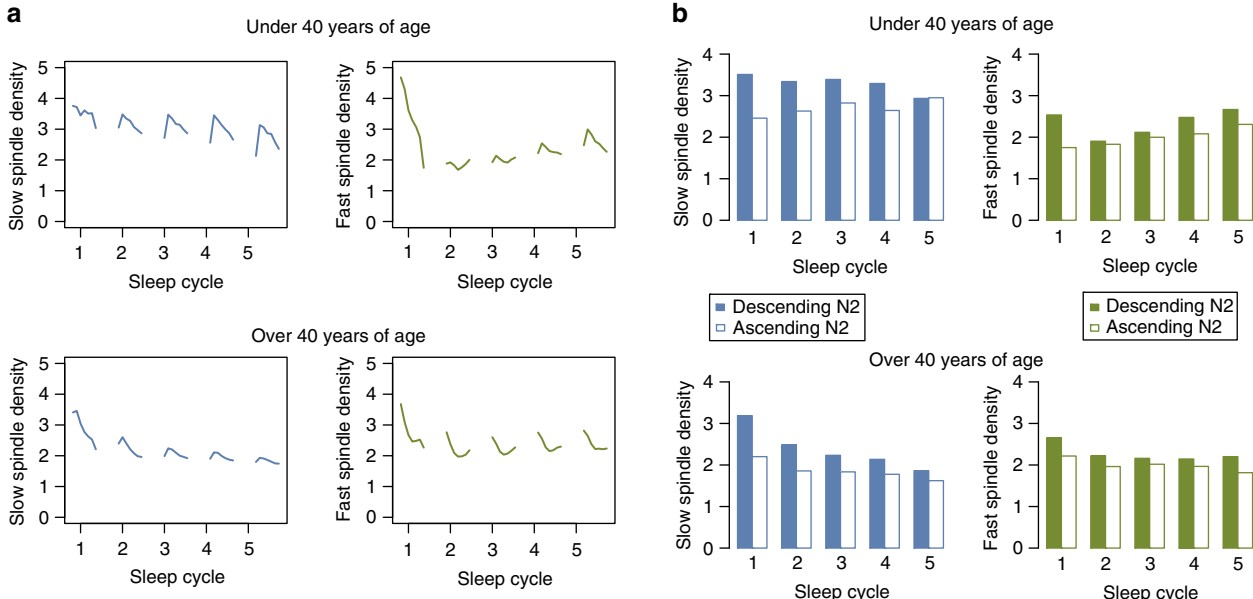

**Figure 3 | Spindle density across the night.** Mean spindle density (spindles per min) during N2 by sleep cycle and position within the cycle, stratified by age (years) and spindle frequency (Hz). (**a**) For slow (11 Hz, blue, left column) and fast (15 Hz, green, right column) spindles, mean density averaged over epochs (rather than individuals) stratified by sleep cycle, separately for individuals under 40 (mostly children and adolescents, mean age ~12 years) versus over 40 (mean age ~68 years). The first 60 min of each cycle was divided into six 10-min intervals of N2 sleep; an additional >60 min point is included for each cycle also. (**b**) Spindle density in ascending (unfilled bars) versus descending (filled bars) N2 epochs, stratified by sleep cycle number, age and spindle frequency.

**Changes in spindle activity within NREM sleep cycles**. Even within N2 sleep, spindle activity was not uniform within a typical sleep cycle, but dependent on local temporal context, namely the relative position within the sleep cycle and the type of sleep or wake that preceded or followed a particular N2 epoch. Dividing the first 60 min of each cycle into six 10-min intervals, and fitting an epoch-level linear mixed model with fixed effects of sleep cycle number (as a five-level factor) and within-cycle position (as a six-level factor), marked systematic within-cycle variation was observed for both fast and slow spindles, although the two types showed different profiles (Fig. 3a and Supplementary Table 12).

Specific features of the hypnogram might account for part of this within-cycle variability. Neural activity during the so-called descending (transitioning from N1, REM or wake to N3 sleep) versus ascending (transitioning from N3 sleep to N1, REM or wake) N2 epochs (see Methods and Supplementary Fig. 15) may systematically vary, reflecting the different direction of travel along the sleep–wake continuum[62]. Indeed, both fast and slow spindles were significantly enriched (Fig. 3b and Supplementary Table 12) during descending N2 sleep (mean epoch-level spindle densities were 2.61 and 2.44 for slow and fast spindles, respectively) compared to ascending N2 sleep (2.15 and 2.14). The remaining N2 epochs (neither clearly ascending nor descending) showed intermediate densities (2.26 and 2.33). Analyses controlled for sleep cycle number, as ascending/descending epochs tended to occur earlier due to the higher rate of N3 sleep during the first half of the night. This effect was observed broadly, across sleep cycles, age groups and spindle frequencies, in analyses of all N2 epochs as well as only during persistent sleep (previous wake being more than 10 min prior).

We also focused on NREM–REM transitions, as studies in rodents have pointed to increased spindle activity immediately preceding REM sleep[63]. Controlling for sleep cycle and other factors including relative position within the sleep cycle, we observed an increase in fast spindle activity preceding a transition from N2 to REM sleep, but a decrease in slow spindles (Supplementary Fig. 16). These changes were relatively gradual, particularly for fast spindles, happening over the course of 5 or more minutes rather than immediately preceding the state change. In contrast, we did not observe marked changes in spindle activity before wake, after controlling for sleep cycle and other factors.

These within-cycle associations were observed whether considering all N2 epochs, or only those during persistent sleep. In general, N2 spindles occurred less often during persistent sleep compared to the 10 min following a wake epoch (Supplementary Table 12), with the effect being particularly strong for fast spindles. Stratifying analyses by sleep cycle, for fast spindles this effect was observed for each cycle, whereas for slow spindles the effect was concentrated in the first and second cycles. As shown in Fig. 3a, spindle density (for fast and slow spindles, in both younger and older individuals) was highest near the start of the first cycle. This effect remained when considering only N2 epochs during persistent sleep that were flanked by at least 4 other epochs of N2 (before and after). Spindle activity following initial sleep onset may be qualitatively different from spindle activity during the rest of the night: we noted that per-epoch spindle density at the start of the night (approximately within the first 5 min) tended to be less correlated with average spindle density over the whole night (Supplementary Fig. 17). Early-night N2 spindles showed no corresponding drop in the test–retest correlations, however (Supplementary Fig. 17), suggesting that the first-cycle peaks in spindle activity evident in Fig. 3 do not simply reflect artefact.

To summarize within-cycle variation in spindles: fast spindles were increased following wake and preceding REM sleep, whereas slow spindles were reduced preceding REM sleep. Both fast and slow spindles showed their highest levels of activity around the start of the first sleep cycle. All cycles show significant within-cycle variability in N2 spindles that depended on age and spindle frequency. Unpacking which of these and other highly

interrelated ultradian factors are causally related to spindle activity remains a challenge, however.

**Epoch-level models of circadian modulation.** Sleep spindles are known to show marked circadian effects[64–66]. Previous reports have noted that fast and slow spindles have 180° phase-shifted circadian rhythms[64], in which slow spindles peak around the time of maximum melatonin levels (typically around 3:00–4:00), whereas fast spindles have their nadir at that point. Furthermore, circadian modulation of spindles has also been shown to be attenuated in older individuals[65,66]. To unambiguously disentangle circadian from sleep homeostatic and other ultradian factors requires forced desynchrony or constant routine experimental paradigms. Nonetheless, we attempted an approximate characterization of circadian modulation, by leveraging the differences between local clock time and elapsed time that arise from the variation in sleep onset across NSRR individuals.

We annotated each epoch with respect to local clock time, elapsed time since lights out, elapsed sleep, elapsed N2, elapsed N3 and elapsed REM. Comparing a series of mixed models that predicted per-epoch fast and slow spindle density from each of these terms, the elapsed-sleep model yielded the best fit (Supplementary Fig. 18a). All models controlled for sleep cycle number and within-cycle effects as well as individual-level fixed effects of age, sex, study and race. That is, minutes of elapsed sleep (allowing for higher-order terms) significantly predicted spindle dynamics over and above the variation captured by between- and within-cycle effects (Supplementary Fig. 18b). Model fit was further improved by adding local clock time in addition to minutes of elapsed sleep, consistent with the proposition that circadian factors (that is, as approximated by variation in clock time that is independent of variation in elapsed sleep) modulate spindle activity on top of sleep homeostatic and ultradian factors. We did not observe any association between sleep mid-point—typically a proxy for chronotype—and average spindle density across the night.

Although clock time, circadian phase and chronotype have a complex relationship, we assume that for sleep onset during the polysomnography, individual differences in clock time at least partially reflect variation in circadian phase at sleep onset, and are not entirely explained by individual differences in chronotype. To characterize the nature of the putative circadian modulation, we therefore made the critical assumption that after accounting for the effects of elapsed sleep on spindle density, any residual covariation with local clock time indexed circadian modulation. Plotting the residuals from the elapsed-sleep model against local clock time (Supplementary Fig. 19), we indeed observed systematic trends across the night that were consistent with the two main features highlighted above: qualitatively different circadian modulation of fast and slow spindles that was present only in younger individuals.

**Medication effects and sleep apnoea.** Medications that affect sleep can also affect spindles, most notably benzodiazepines and certain non-benzodiazepine sedative hypnotics[67]. Information on medication use was available for two studies. In SHHS, benzodiazepines were the drugs most strongly associated with increased spindle density ($P = 4 \times 10^{-5}$, 0.2 more spindles per minute for the $N = 305$ of 5,793 individuals taking benzodiazepines, linear regression of spindle density on medication status and standard covariates; Supplementary Table 13 and Supplementary Fig. 20). Based on a similar analysis in MrOS, benzodiazepine use was also associated, with 0.27 more spindles per minute in the 133 of 2,907 individuals taking this medication ($P = 0.0003$; Supplementary Table 14). The most highly

associated medication in MrOS was zolpidem ($P = 1 \times 10^{-5}$, 0.58 more spindles per minute in $N = 40$ individuals), a non-benzodiazepine sedative hypnotic that binds to GABA (γ-aminobutyric acid) receptors at the same location as benzodiazepines.

A non-trivial proportion of NSRR individuals were recruited for sleep apnoea symptoms (Supplementary Table 15). The 12% of the sample with severe apnoea (AHI >30) had significantly lower spindle densities, after controlling for age, sex, race and study ($P = 3 \times 10^{-5}$, $N = 10,023$, with 0.12 fewer spindles per minute compared to individuals without sleep apnoea/with low AHI). Treated as continuous measures, AHI and also the arousal index were both moderately associated with reduced spindle density (Supplementary Table 16), although these associations were weaker than those for other measures, including age. Nevertheless, we elected to include arousal index and AHI as covariates in all primary analyses.

**Other demographic correlates.** As previous studies have reported racial differences in sleep architecture[68], we characterized group-level associations between race and spindle activity. Across the six studies 2,575 individuals reported a non-white racial identity, the majority of whom ($N = 1,815$) identified as black (Table 1). Compared to whites, blacks spent more time in N2 and less time in N3, and also tended to have more fragmented sleep (Supplementary Tables 17 and 18).

Within N2 sleep, blacks tended to show lower absolute and relative measures of spectral power, particularly in the sigma range (Supplementary Fig. 21 and Supplementary Table 19), with or without adjusting for age, sex and other potential confounders. Statistically significant across the entire sample, this effect was nonetheless concentrated in younger individuals, with the older studies (MrOS and Study of Osteoporotic Fractures (SOF)) not showing any group differences. This did not simply reflect lower power due to the smaller proportion of black individuals in MrOS and SOF (Table 1): a significant age-by-race interaction in the entire sample implied a reduced effect of race in older individuals ($P < 10^{-15}$, $N = 9,747$, linear regression on sigma power).

Blacks had significantly lower estimates of spindle density (0.3 per minute), with the effect also concentrated in younger individuals (Supplementary Fig. 22 and Supplementary Table 20) and a significant age-by-race interaction consistent with this ($P = 2 \times 10^{-7}$, $N = 9,820$, linear regression on spindle density). Blacks also showed lower spindle amplitudes and shorter spindle durations, but no differences in spindle frequency (Supplementary Table 21). In the frequency-dependent spindle analysis, the association with race was observed across a range of spindle frequencies. Supplementary Fig. 23 summarizes the age-dependent racial differences in spindle density and duration of N3 sleep. These two associations were independent of each other, in that controlling for one variable did not diminish the association between race and the other variable.

Although spindle activity is influenced by genetic factors, there are clearly major non-heritable sources of variation including age, sex and medication use. Similarly, although group differences between races could in part be driven by genetic factors, they could equally arise due to unmeasured environmental confounding factors. Understanding the causes (and consequences) of any racial differences remains an open empirical question. Many factors that we were unable to capture comprehensively, including socioeconomic status (SES) at both family and neighbourhood levels, environmental noise and pollution, diet and normal sleep habits, could be causal factors that mediate the statistical association with race. For four studies, indices of family SES were available, including a self-reported ladder measure, household income, and parental education and employment. All

SES measures were significantly lower in blacks compared to whites, and some were modestly associated with spindle density (Supplementary Table 22). However, in joint models predicting spindle density as a function of race, SES and other covariates, none of the SES indices remained significant, whereas race continued to be a moderately strong predictor of spindle density. Thus, the available individual or family-level indices of SES were not better proxies to describe the causal nature of the observed statistical association between race and spindle activity. Further research to more fully characterize individual differences in brain activity during sleep should strive to identify the proximal, causal factors that underlie these apparent group-level racial differences.

**Heritability of sleep spindles.** We leveraged the family-based CFS to test for genetic influences on spindle measures[16–18], as exemplified by two monozygotic twin pairs in that study. In comparison with two other randomly selected unrelated pairs (but matched for age, sex and race), the greater concordance between MZ twins is evident for both spectral and spindle traits (Fig. 4 and Supplementary Table 23).

We used available genome-wide microarray single-nucleotide polymorphism (SNP) data to estimate additive trait heritabilities ($h^2$) and genetic correlations ($r_G$), using variance components models[69]. Separately for blacks and whites, and excluding the two MZ twin pairs, we consistently observed evidence of genetic influences on spindle density ($h^2 = 0.45$, $P = 8 \times 10^{-6}$ for $N = 186$ whites; $h^2 = 0.43$, $P = 3 \times 10^{-6}$ for $N = 229$ blacks) and other sleep architecture, spectral and spindle traits including sigma power (Table 2). A different approach not based on SNP data (intraclass correlations (ICCs) for full sibships) similarly found sibling correlations (ICC $\sim 0.2$–$0.4$, correcting for age, sex and race effects) for all spectral and spindle traits, including sigma power, spindle density and spindle amplitude, consistent with substantial genetic contributions (Supplementary Table 24).

A genetic correlation reflects the extent to which two traits are influenced by the same sets of genetic variants. Canonical spindle density and amplitude were positively genetically correlated ($r_G = 0.49$ and $0.43$ for whites and blacks, respectively, both $P < 0.05$, $N$ as above; Supplementary Table 25). Spindle density

also had high genetic correlations with duration of N2 ($r_G = 0.62$ and $0.93$ for white and black individuals, respectively), and negative genetic correlations ($r_G = -0.45$ and $-0.48$) with duration of N3, suggesting that the genes that affect spindle density also influence NREM duration. Despite the modest genetic correlations with absolute sigma power shown in Supplementary Table 25, spindle density was highly genetically correlated with relative sigma power ($r_G = 0.69$ and $0.68$ in whites and blacks, respectively, both $P = 0.0001$, not shown in the table). In contrast, we observed negative phenotypic and genetic correlations between spindle density and delta power. There were very high genetic correlations for estimates of spindle density in N2 and N3 ($r_G = 0.89$ and $0.88$ in whites and blacks, respectively), suggesting that similar genetic variation influences density across both stages.

Despite significant univariate heritability for spindle frequency, there was no evidence for genetic overlap with spindle density, suggesting that genes that influence how often spindles occur tend not to influence their frequency, and *vice versa*. Others have speculated that different mechanisms may be responsible for changes in spindle frequency versus spindle density, for example, changes in thresholds for $Ca^{2+}$ spikes versus impairments in recruitment mechanisms or internal desynchronization of neurons in the thalamic nucleus[39]. Considering fast and slow spindle density separately, there was no evidence for genetic overlap in densities ($P = 0.31$ and $0.18$ for whites and blacks, respectively) despite both showing significant heritability (Supplementary Table 26). Figure 5 shows the univariate and bivariate genetic parameters for the densities across the full range of targeted frequencies ($F_C = 8$ to $18\,Hz$). In general, faster spindles ($F_C \sim 15\,Hz$) appeared to have greater heritability, with a second peak around slower spindles ($F_C \sim 11\,Hz$), but with no evidence for genetic overlap between these two types.

**Topographical analyses.** There was substantial spectral coherence between C3 and C4, with differences in sigma-band coherence in part driven by (or at least correlated with) differences in spindle activity (Supplementary Table 27). We observed subtle but statistically significant (paired $t$-test, $P < 10^{-15}$,

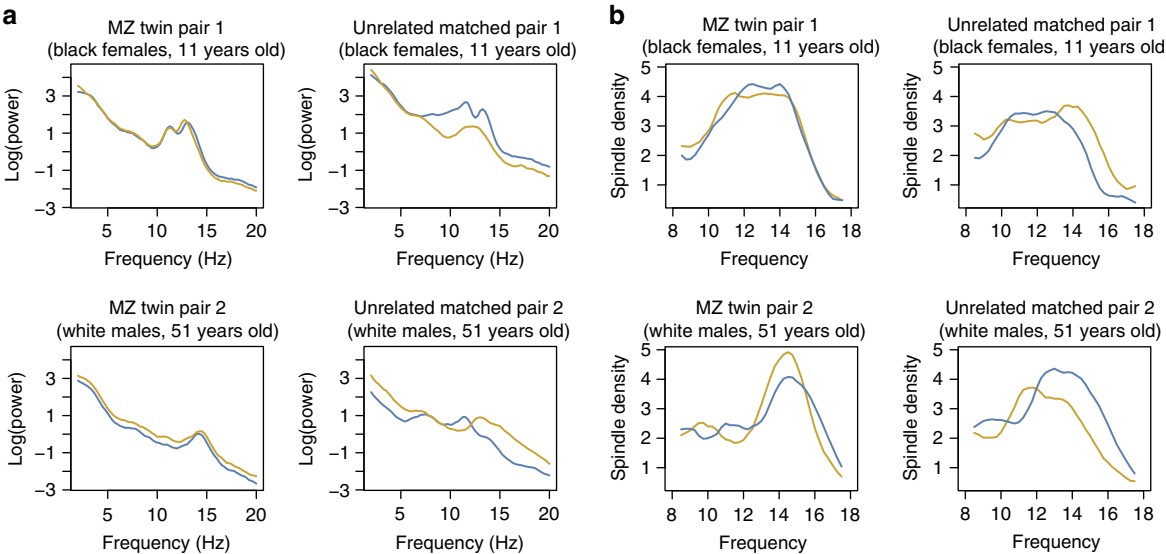

**Figure 4 | Spectral and spindle traits for two MZ twin pairs and two matched unrelated pairs.** Data from two MZ twin pairs in the Cleveland Family Study, and two randomly selected unrelated individuals (matched for age, sex and race to the corresponding MZ pair, see Supplementary Table 23). (**a**) Spectral power for each pair; in each plot, different coloured lines represent the first and second members of each pair. (**b**) Similar results for spindle density from the frequency-dependent spindle analysis ($F_C$ varied from 8 to 18 Hz). In all cases, the MZ pairs showed greater concordance in spectral and spindle traits.

**Table 2 | Univariate heritability estimates for sleep and spindle traits.**

| Phenotype | White CFS families | | Black CFS families | |
|---|---|---|---|---|
| | $h^2$ | P-value | $h^2$ | P-value |
| *Sleep stage duration (min)* | | | | |
| N1 sleep | 0.26 | 0.002 | 0.40 | $2 \times 10^{-5}$ |
| N2 sleep | 0.21 | 0.03 | 0.06 | 0.27 |
| N3 sleep | 0.43 | $1 \times 10^{-5}$ | 0.22 | 0.008 |
| REM sleep | 0.20 | 0.003 | 0.19 | 0.016 |
| Total sleep time | 0.21 | 0.01 | 0.24 | 0.009 |
| | | | | |
| *Spectral band power* | | | | |
| Slow | 0.26 | 0.008 | 0.20 | 0.006 |
| Delta | 0.26 | 0.004 | 0.28 | 0.0001 |
| Theta | 0.29 | 0.009 | 0.30 | $3 \times 10^{-5}$ |
| Alpha | 0.41 | 0.0002 | 0.34 | $2 \times 10^{-6}$ |
| Sigma | 0.74 | $3 \times 10^{-10}$ | 0.49 | $2 \times 10^{-8}$ |
| Beta | 0.43 | 0.0005 | 0.43 | $4 \times 10^{-8}$ |
| | | | | |
| *Spindle traits (N2 sleep)* | | | | |
| Density | 0.45 | $8 \times 10^{-6}$ | 0.43 | $3 \times 10^{-6}$ |
| Amplitude | 0.48 | $7 \times 10^{-6}$ | 0.39 | $3 \times 10^{-6}$ |
| Duration | 0.39 | 0.0003 | 0.23 | 0.01 |
| Frequency | 0.39 | 0.0008 | 0.33 | 0.0003 |
| Oscillations | 0.41 | 0.0002 | 0.20 | 0.02 |
| Symmetry | 0.31 | 0.001 | 0.11 | 0.10 |

CFS, Cleveland Family Study; REM, rapid eye movement; SNP, single-nucleotide polymorphism. Separately for black and white individuals from the CFS, heritability ($h^2$) was estimated using mixed models applied to SNP microarray data. Owing to the relatively small sample sizes for this type of analysis, confidences intervals will be broad and so stochastic fluctuations in reported $h^2$ values are to be expected, and should not be overinterpreted. Most aspects of sleep architecture (minutes in N1, N3 and REM) showed significant heritability. Spectral band power for all frequencies showed significant genetic influence, in particular sigma activity. (Similar results were obtained using relative rather than absolute band power, data not shown.) Spindle traits during N2, in particular density and amplitude, also showed highly significant ($P < 10^{-5}$) estimates for heritability, ~0.4–0.5. Intraclass correlations based on first siblings were also consistent with heritable influences on spindle traits (Supplementary Table 24).

$N = 11,035$) mean differences between left (C3) and right (C4): spindles at C3 tended to occur at a 1.5% higher rate and be 3.2% stronger. Approximately half of all spindles were bilateral (detected at both C3 and C4 within 0.5 s of each other), although this rate decreased with increasing age (Supplementary Table 27). These effects could be confounded with spindle density and amplitude: having more spindles increases chance overlap and higher amplitude spindles may be more likely to be detected bilaterally due to increased volume conductance. Controlling for spindle density, amplitude and other covariates, however, the association with age remained, consistent with reports of topographical changes in spindles over development[13].

A subset of the CHAT study ($N = 53$ children) had EEG data on 18 scalp electrodes. Consistent with previous reports[70], slower ($F_C = 11$ Hz) spindles were significantly enriched at frontal channels (F3 and F4) compared to C3, whereas faster ($F_C = 15$ Hz) spindles exhibited a more uniform topography (Fig. 6a,b). Because future genetic studies on NSRR data will typically only have EEG from central electrodes, we characterized the relationship between within-channel variability in spindle density at C3/C4 compared to that at other locations, which will not be directly assayed. Considering the correlational structure of individual mean spindle densities, spindle frequency was a larger source of variability than topography (Supplementary Fig. 24). That is, detecting slow spindles at C3 or C4 captured most of the within-channel variance in slow spindles at other locations (Fig. 6c and Supplementary Fig. 25). Similarly, with the exception of temporal and occipital channels with very low mean spindle densities, fast spindle density at C3 and C4 was highly predictive of fast spindle density globally (Fig. 6d and Supplementary Fig. 25). Although more nuanced topographical questions (for example, of connectivity, relative topographic profiles or spindle propagation) will be impossible in studies with only one or two central electrodes, these results suggest that such studies

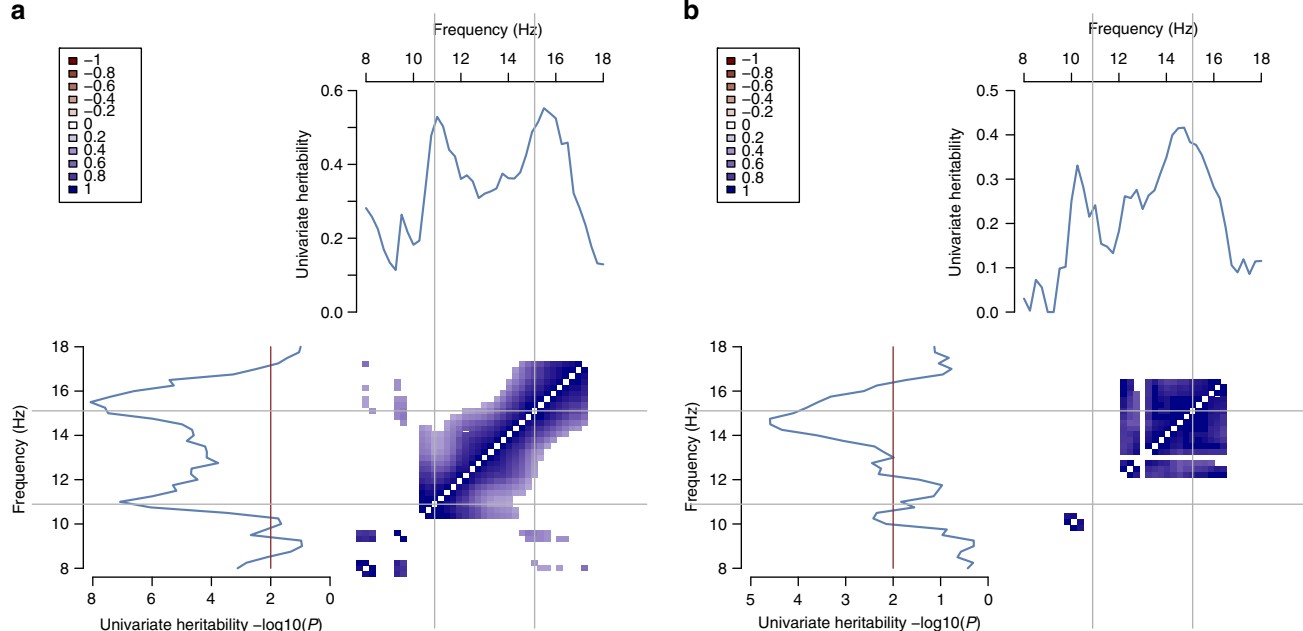

**Figure 5 | Heritabilities and genetic correlations for spindle density across a range of frequencies.** The plots show the estimated heritability for spindle density at a range of $F_C$ (8–18 Hz) from the frequency-dependent spindle analyses. Heritabilities were estimated by applying mixed models to SNP data available in the CFS, restricted to either (**a**) black or (**b**) white individuals. In each panel, the upper right plot gives the estimated heritability ($h^2$) and the lower left plot gives the corresponding significance value for the test of H$_0$: $h^2 = 0$. Shaded points in the lower right quadrant represent genetic correlations ($r_G$) between spindle densities at different $F_C$, only showing significant ($p < 0.05$) correlations. Grey bars highlight $F_C = 11$ and 15 Hz, which correspond to 'slow' and 'fast' spindles. Although both types of spindle showed significant univariate heritability, there was little evidence for shared genetic factors (significant $r_G$).

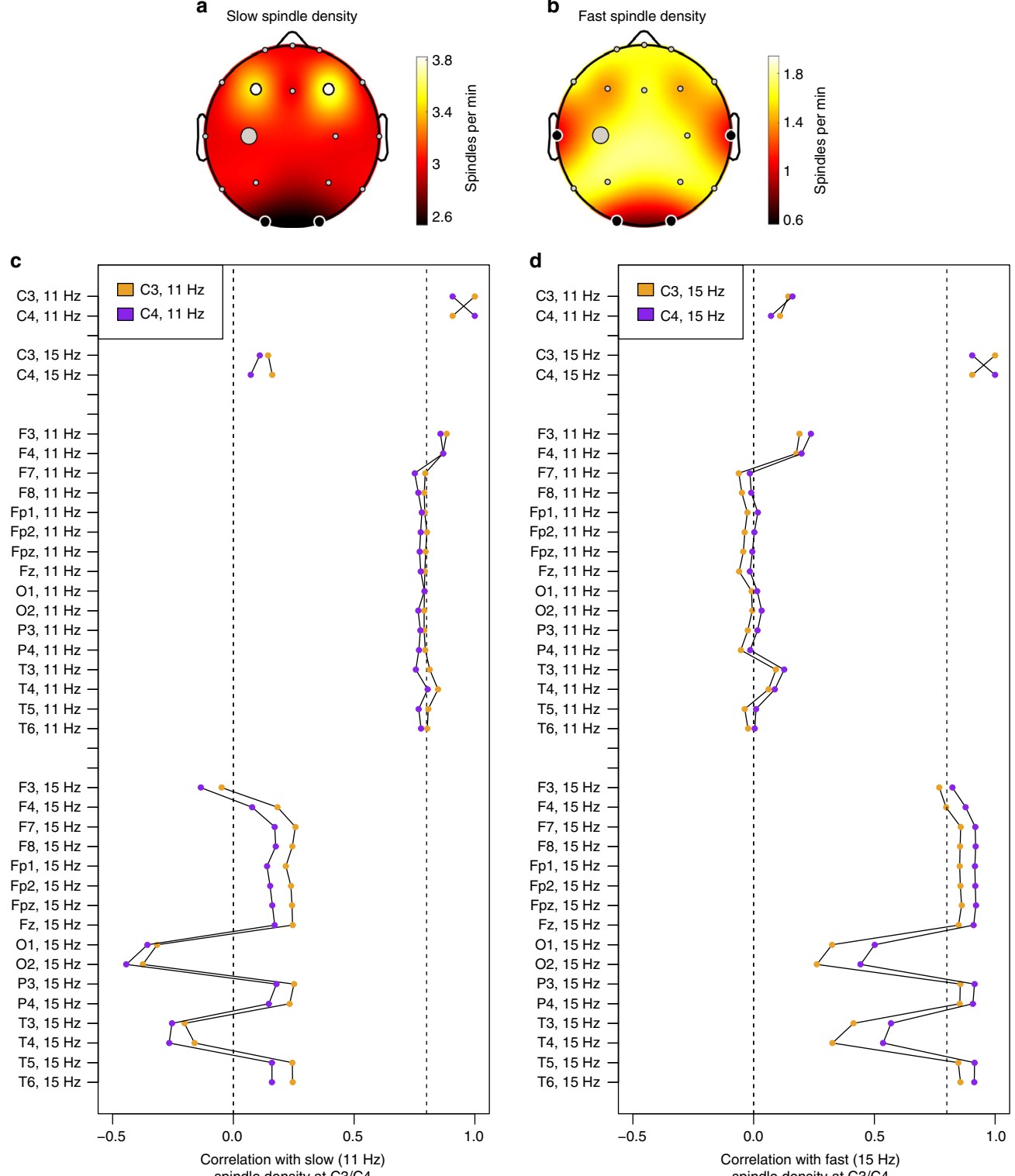

**Figure 6 | Topographical analyses of individual differences in spindle density in the CHAT study.** (**a**) Mean slow ($F_C = 11$ Hz) spindle densities for the subset of individuals in the CHAT study with EEG at 18 electrodes. Electrodes with means significantly ($P < 0.05/17$, that is, Bonferroni correction for 17 tests) greater (or less) than C3 (grey circle) are shown as enlarged white (or black) circles. (**b**) As above, but for fast ($F_C = 15$ Hz) spindles. (**c**) Correlation in individual slow spindle densities at C3 and C4 across other locations, for both fast and slow spindles. For reference, dotted lines represent $r = 0.0$ and .8. (**d**) As above, but for fast spindles. Supplementary Fig. 25 shows the underlying scatter plots for each correlation point here.

will nonetheless be able to capture most of the within-channel variation in spindle activity between individuals. Fast and slow spindle density estimates showed little or no correlation with each other, either within the same channel or across different channels.

**Spindle detectability and measurement of spindle parameters.** To detect spindles, we used a previously published wavelet algorithm[21] that has been comprehensively and independently evaluated[38]. To establish the robustness of our findings we also

adopted a second algorithm, based on thresholding the signal's root mean square value after band-pass filtering in the sigma range[20]. We observed a high correlation (Pearson's $r = 0.83$, $P < 10^{-15}$, $N = 11,088$) between estimates of spindle density across the two methods (with a tendency for higher correlations in the older samples, likely reflecting the shift in spindle frequency that occurs with age), although the band-pass method detected more spindles, with a mean of 2.97 spindles per minute, compared to 1.88 (Supplementary Table 28). The wavelet method yielded estimates of spindle density that tended (1) to be more consistent over time within an individual, (2) had stronger associations with the known demographic factors of age and sex and (3) showed higher sibling intraclass correlation (Supplementary Table 29). However, our primary, substantive results did not depend on which method was used.

To extract discrete events from the continuously varying EEG, spindle detection methods typically set thresholds for duration, frequency or amplitude, although these can be arbitrary and not based on any biological ground truth, whether they are fixed (for example, using the same μV amplitude threshold across all individuals) or adaptive (for example, empirically determined for each individual relative to background sigma activity). Our primary analyses used the default, previously published value ($t = 4.5$ times the mean[21,38]) for an adaptive amplitude detection threshold. We assessed sensitivity by considering putative spindles only detected at lower amplitude thresholds: low-amplitude spindles ($T_L$) detected at $1 < x < 2$ and all spindles ($T_A$) detected at $x > 1$, compared to the standard ($x > 4.5$) threshold, where $x$ is the wavelet coefficient divided by that individual's mean for all N2.

Visual inspection underscores the necessarily subtle nature of these more obscure spindles, which would be unlikely to be flagged by human raters (Supplementary Fig. 26). Although lower than the standard, estimates of spindle density based on $T_L$ and $T_A$ nonetheless exhibited moderately high test–retest correlations (Supplementary Table 30), suggesting that they measure more than stochastic noise and pointing to an obvious tradeoff between sensitivity and specificity. More importantly, we observed a substantial but negative correlation (Pearson's $r = -0.55$, $P < 10^{-15}$, $N = 9,944$) between standard and $T_L$ estimates of spindle density: individuals with more high-amplitude spindles (satisfying the default threshold) tended to have fewer low-amplitude ones, and *vice versa*. Consistent with this, $T_L$ spindles showed highly significant associations that were in opposite directions compared to the standard analysis, for variables including benzodiazepine use (Supplementary Table 31), as well as age, sex and race (Supplementary Table 32). Similarly, the life-course trajectories for low-amplitude $T_L$ spindles were near mirror images of those for higher amplitude spindles (Supplementary Fig. 27).

One possible explanation for these results is that spindle amplitude rather than density is more fundamentally related to these factors. If the probability of detecting a spindle is not independent of its amplitude, then, depending on the sensitivity of the detector, true changes in amplitude will appear as changes in estimated spindle density. Supplementary Fig. 28 posits an illustrative model in which an increase in true spindle amplitude leads to an increase in the estimated density of high-amplitude spindles, but a decrease for low-amplitude spindles, in the absence of any true changes in spindle density. Consistent with this model, when revisiting the association between spindle density and race, we found that spindle amplitude was the only metric—over and above spindle density and background sigma power—that showed significant and independent effects (Supplementary Table 33).

## Discussion

In summary, here we detect and characterize spindle activity in 11,630 individuals using an automated pipeline. The resulting estimates of spindle density, amplitude, duration and frequency are heritable, stable traits amenable to future genetic analysis. Our findings point to phenotypic and genetic heterogeneity, as we find qualitatively different profiles for fast and slow frequency spindles with respect to age and sex, topography and dynamics across the night. We further show that fast and slow spindles are effectively independent at the population level and are not genetically correlated, suggesting relatively distinct aetiologies and functions. Although we use the terms fast and slow as a shorthand, our results do not directly and unambiguously support the existence of two discrete spindle types, however. It is still unclear whether there are two or more types of spindles, or whether a continuous spectrum of faster and slower spindles provides a better characterization.

We identified potential confounding factors, underscoring the importance of careful matching on race and appropriate modelling of age, sex and other effects in studies that use spindle activity as a biomarker of disease, or in molecular genetic studies. Whether any one of these associations reflects true biological variation in neural activity, versus confounding by other latent factors, is unclear. For example, although not necessarily a major determinant of individual differences in the EEG[71], morphological features such as skull thickness could in theory influence spindle measurement.

Although we framed our primary analyses in terms of spindle density, which is typically the focus of other studies, we showed that differences in estimated spindle density between individuals or conditions could reflect differences in spindle amplitude, or other factors which influence detectability such as duration, frequency or topography. Analysis of density alone could conceivably mask important biological distinctions, as the circuitry controlling the onset of spindles, versus their oscillatory behaviour, versus their amplitude, could be largely distinct. Whether or not spindle density and amplitude are truly correlated across individuals (as opposed to this correlation being artificially induced by the choice of amplitude threshold) is also an open empirical question. Furthermore, learning-induced state changes in spindle activity may reflect true changes in density rather than amplitude. Moving forward, analyses focused on the density of high-amplitude spindles (that is, our default analysis) should be supplanted by more exact methods for parsing spindle density and spindle amplitude as separable components.

A number of other caveats are warranted. Automated detectors (see ref. 72 for an excellent review) are clearly not optimal, although human scorers also exhibit less than perfect inter-rater reliabilities[38]. Furthermore, the $T_L$ and stage N3 analyses point to spindle activity that would be unlikely to be detected visually, but nonetheless may be valid. Perhaps, the most pernicious measurement issue (for both human and automated scores) is the dependency between the true properties of a spindle and its detectability.

A limitation of the present study is the under-representation of young adults, as it precludes more fine-grained comparisons between adolescents and young adults, as well as a higher-than-average proportion of individuals with sleep apnoea symptoms. The inability to fully characterize the topographic heterogeneity in the full sample of adults is a further limitation. We are also unable to consider comprehensively and directly other factors that have been shown to influence spindles, including menstrual cycle and circadian modulation[65]. We present an approach to infer circadian modulation of spindles indirectly which, although far from the gold standard, may have potential for scoring individuals with respect to their magnitude or type of circadian

modulation. For example, the cross-covariance of elapsed-sleep model residuals with a known or inferred circadian signature could yield traits that could be associated with genotype or disease state. An exact partitioning of circadian and non-circadian dynamics is beyond the scope of this approach, which makes the strong assumptions that (1) a sufficiently accurate model of sleep homeostatic and ultradian factors can be deployed, and (2) that local clock (or zeitgeber) time is a sufficient proxy for circadian phase, despite the subtleties in the relationship between circadian phase, clock time and chronotype[73].

Finally, it is important to stress that in a large sample size, highly significant associations do not necessarily equate to large, interesting or direct, causal effects. We hope that, in the future, molecular genetic studies will help to unravel the causal nature of the associations between spindles, spectral measures, other aspects of sleep and biomedically relevant traits and diseases.

## Methods

**Overview of National Sleep Research Resource data.** All data were obtained from the NSRR (http://sleepdata.org), including EEG and electrocardiogram (ECG) signals on 11,630 individuals across six studies (Table 1, Supplementary Fig. 1, Supplementary Methods and Supplementary Note 1). All data were collected as part of research protocols that were approved by the local institutional review board at each institution; written, informed consent was obtained from each individual before participation. The majority of individuals were from community-based samples, although two studies recruited participants for sleep apnoea. A subset ($N = 4,079$) had a second polysomnogram, typically administered 5 or 6 years after the first. European Data Format (EDF) files and annotation files (indicating manually scored sleep stages in 30-s epochs, and manual annotations for arousals, limb movements and signal artefacts) were accessed on January 2016. Electrode labels are from the International 10–20 system. All studies used American Academy of Sleep Medicine staging, except the SHHS, which used R&K: here NREM3 and NREM4 were collapsed to a single N3 stage, for consistency with the other studies.

**Description of studies.** Individuals ranged in age from 4 to 97 years and were well balanced between males and females. In tabulating results, studies are ordered by average age: the CHAT (children), the Cleveland Children's Sleep and Health Study (CCSHS, adolescents), the CFS (predominantly adolescents and middle-aged adults), the SHHS (middle-aged adults), the MrOS (elderly males) and the SOF (elderly females). We stratified analyses by study to control for possible technical and measurement differences as well as the effects of ageing.

Key covariates for all individuals included age (years), sex and racial/ethnic status (coded black, white and other). Unless explicitly noted otherwise, the three levels of the race factor are modelled as two dummy-coded binary variables (typically when race is included as a covariate). Comparisons involving race but that indicate only the levels 'black' or 'white' (typically where race is the focus of the analysis) imply a comparison of that group against the other, that is, excluding individuals coded as 'other' due to relatively low numbers. Individuals' sleep was scored for an arousal index (the total number of arousals per hour of sleep) and AHI (the overall respiratory disturbance index at 3% oxygen desaturation).

Unless explicitly noted otherwise, all primary individual-level analyses control for age, sex, race, study, arousal index and AHI (entered as covariates in linear regression models). Unless noted otherwise, analyses are based on the set of independent observations from the first polysomnogram ($N = 11,630$). Table 1 gives $N$'s for individual studies; some specific tests will have slightly fewer observations because of small amounts of missing data. All $P$ values are two sided; those smaller than $10^{-15}$ are reported as $P < 10^{-15}$. The total sample size means we are well powered to replicate existing findings from studies two or three orders of magnitude smaller, as well as considering heterogeneity between and within studies.

**Signal artefact detection.** For all EDFs, we extracted signals for the two central derivations (C3-A2, C4-A1) and, where available, a single ECG channel (Supplementary Fig. 2). For the primary analyses, only epochs annotated as N2 were retained. Any epoch with an overlapping arousal, movement or signal artefact annotation was removed from analysis. EEG and ECG signals were resampled at 128Hz (with a small number of exceptions, the original EEG sampling rates were as follows: CHAT: 200, 256 or 512 Hz; CCSHS: 128 Hz; CFS: 128 Hz; SHHS: 125 Hz; MrOS: 256 Hz; SOF: 128 Hz; the initial ECG sampling rates were as follows: CHAT: 200, 256 or 512 Hz; CCSHS: 256 Hz; CFS: 128, 256 or 512 Hz; SHHS: 125 Hz; MrOS: 512 Hz; SOF: 256 Hz). EEG signals were filtered with a zero-phase band-pass filter (0.3–35 Hz). Aberrant epochs were removed following the procedure described in ref. 8. Briefly, for each 30-s epoch, we calculated spectral band power for delta- (1–4 Hz) and beta- (15–30 Hz) bands, using fast Fourier

transformation and the Welch algorithm (4-s sliding window with 50% overlap). Comparing each epoch to the moving average based on up to 15 contiguous epochs (7 either side), an epoch was excluded if the delta power was > 2.5 times the local average, or if the beta power was > 2.0 times the local average. Epochs flagged as aberrant for either channel were removed from all analyses downstream, so for a given individual, analyses were based on the same set of epochs for each signal.

The above Buckelmüeller et al.[8] filtering procedure sometimes failed to remove a significant number of clearly aberrant epochs. We therefore conservatively applied an iterative series of statistical filters to four per-epoch summary metrics, removing any epoch that was ± 2 s.d. units from the mean value for that individual, on either EEG channel. The four metrics used were the root mean square and three Hjorth parameters (activity, mobility and complexity)[74]. Outlier detection was iteratively performed three times because some signals contained a large number of extreme outlying epochs, which inflated the total variance leading to non-trivial numbers of less extreme but still aberrant epochs being retained after a single round of filtering. We also removed any epoch for which more than 5% of the sample points were tied at the minimum or maximum value (that is, clipped signals).

**Spindle detection.** The primary method was based on the Morlet wavelet transformation[21], defined as

$$\psi(x) = (\pi F_B)^{-0.5} \exp(2\pi i F_C x) \exp(-x^2/F_B)$$

where $F_B = 2s^2$ and $s = n/2\pi F_C$, where $n$ is the number of cycles of the complex Morlet wavelet. $F_C$ is the centre frequency, the frequency targeted most strongly, set to $F_C = 13.5$ Hz in the canonical analyses. We secondarily varied $F_C$ from 8 to 18 Hz in 0.25 Hz increments (here labelled the frequency-dependent analysis), to target spindles across a broader range of frequencies. $F_B$, the wavelet bandwidth, is determined by the number of cycles ($n$) for a given $F_C$. Varying $n$ represents a tradeoff between resolution in the frequency versus the time domain; for the primary analysis, we set $n = 7$, a common default that provides a reasonable balance of time and frequency domain resolution[75]. For the frequency-dependent analyses, we set $n = 12$ to give better frequency resolution.

Wavelet coefficients were smoothed using a moving average (window duration 0.1 s). In the default analysis, sample points with coefficients greater than a multiplicative threshold ($t = 4.5$ times greater than the baseline, which is the average value across all artefact-free N2 sleep for that individual/channel) were flagged. Intervals of consecutively flagged points with durations between 0.3 and 3.0 s were labelled as putative spindle cores. As a modification to the previously published approach, to target the waxing and waning profile of a typical spindle, we further required that all points extending out from each core were greater than a lower threshold of $t = 2$ times the baseline, such that the extended spindle had a total duration of at least 0.5 s. That is, all detected spindles were at least 0.5 s with the signal above $t = 2$ times baseline, and contained a central core of at least 0.3 s with the signal above $t = 4.5$ times baseline. Spindles within 1 s of each other were merged into a single spindle (but only if the resultant merged spindle was < 3.0 s).

As well as the duration (interval of flagged sample points), for each spindle we calculated the frequency, amplitude, number of oscillations and an index of symmetry. Based on a band-pass-filtered signal (11–15 Hz), spindle amplitude was calculated as the largest peak-to-peak amplitude; spindle frequency was calculated as the modal frequency from a fast Fourier transformation; number of oscillations was based on counting peaks and troughs; symmetry was based on the location of the maximum peak-to-peak change relative to the start (0.0) and end (1.0) of the spindle interval.

**Spindle dispersion index.** Clearly spurious spindles were often temporally clustered (Supplementary Fig. 3), reflecting periods of movement or faulty/detached electrodes, leading to a high number of detected spindles in a handful of epochs, but with few or no spindles detected elsewhere (because the baseline for detection has been raised by the artefact). We flagged recordings that exhibited heterogeneous mixtures of N2 epochs with respect to spindle density, taken to be indicative of unreliable detection. Assuming the per-epoch distribution of spindle counts to be approximately Poisson-distributed, for each individual and EEG channel, we calculated the dispersion index (ratio of variance to mean) and tested for overdispersion based on $\chi^2$ goodness of fit. True spindle density fluctuates over the night, so mild overdispersion is expected (observed mean dispersion index was 1.14, significantly greater than the expectation of 1.0 for perfectly Poisson-distributed counts). The extreme tail (values up to 12) more likely reflects artefactual, non-biological sources of variation, however. A threshold of 2.0 was set, which excluded ~1% of signals. (In most cases, the second EEG channel for that individual still provided good data.)

**Detecting and correcting cardiac interference in the EEG.** In preliminary analyses, we noted an unexpected negative correlation (Pearson's $r = -0.15$, $P < 10^{-15}$, $N = 11,142$) between spindle density and body mass index (BMI). Both height (positively) and weight (negatively) were independently associated with spindle density; neck and hip circumference were also negatively associated. Although AHI was also negatively associated with spindle density, the BMI association remained after statistically controlling for age, sex, race, study, arousal

index and AHI in a multiple linear regression of spindle density on BMI and these covariates ($b = -0.0088$ change in spindle density per unit BMI, $P = 2 \times 10^{-8}$, $N = 10,400$). The effect, observed for both wavelet and band-pass detectors, in fact represented greater cardiac interference in the EEG signal in higher BMI individuals, with greater levels of ECG artefact leading to higher thresholds for detecting spindles, and so correspondingly lower rates of spindles.

Cardiac activity can differentially interfere with the EEG as a function of body type, and is often exacerbated in infants and, as we presumed here, more obese individuals. We hypothesized that the spindle–BMI association reflected differential interference of cardiac activity in the EEG. Supplementary Fig. 6a shows the marked differences in EEG/ECG coherence as a function of BMI, present across a range of frequencies including the sigma band. The SHHS had the highest levels of EEG/ECG sigma-band coherence (mean $C_S = 0.13$), although individual $C_S$ correlated with BMI in five of the six NSRR studies examined here (Supplementary Table 1, with four studies in the range of $r = 0.3$–$0.4$). Importantly, this effect was not driven by only a handful of individuals with obvious contamination effects in the EEG.

Using available ECG signals, we effectively removed this cardiac interference (Supplementary Figs 4 and 5), eliminating the marked association with BMI. Cardiac interference in the EEG likely reduced the number of detected spindles because the typical duration of a QRS complex in the ECG is 0.06–0.1 s, approximately in the sigma range (that is, $1/0.06 = 16.6$ Hz). A greater baseline of QRS activity will elevate baseline sigma levels, thereby artificially increasing the threshold above which spindles are detected, leading to a reduced rate of detected spindles. In an attempt to eliminate this effect, we sought to remove the cardiac interference component of the EEG, before spindle detection, using an ensemble average subtraction approach. Specifically, we adopted a modified Pan-Tompkins[76] algorithm to detect R peaks in the ECG (following the implementation here: http://www.robots.ox.ac.uk/~gari/CODE/ECGtools/ecgBag/rpeakdetect.m). Based on filtered N2 sleep, we averaged over intervals of the EEG synchronized by R peaks, up to 2 s in duration, to create a characteristic signature per individual per EEG channel. The signature shown at the top of the Supplementary Fig. 4 is normalized such that each time-point is scaled by the number of data points contributing to that mean, which reduces the stochastic noise at the rightmost end of the signature. For each individual/channel, this average signature was aligned with each R peak and subtracted from the EEG. Epochs with unlikely estimated values for the sleeping heart rate ($<40$, $>100$) were excluded. Fifty-one individuals were missing ECG data: uncorrected values were retained for these individuals.

After the correction, estimates of spindle density were significantly higher (all $P < 10^{-15}$) in all studies (Supplementary Table 1). Across all studies within the NSRR, pre- and postcorrection estimates of spindle density were very highly correlated (Pearson's $r = 0.97$, $P < 10^{-15}$, $N = 11,235$). On average, spindle density was significantly higher postcorrection (1.88 versus 1.72, paired $t$-test, $P < 10^{-15}$, $N = 11,235$). Importantly, however, after ECG correction the association between spindle density and BMI was eliminated in the entire sample (from $P = 2 \times 10^{-8}$, $N = 10,400$ before correction, to $P = 0.25$, $N = 10,438$ after correction, linear regressions on spindle density with standard covariates). A similar pattern was observed for the other anthropometric correlations noted above. Unless otherwise noted, all subsequent analyses were based on the ECG-corrected signals.

Attenuated estimates of spindle activity arising from cardiac interference were also observed in studies outside the NSRR, such as the publicly available DREAMS (http://www.tcts.fpms.ac.be/~devuyst/Databases/DatabaseSpindles/) data set (Supplementary Fig. 6b), suggesting that this is likely a general phenomenon that can impact different studies and analytic approaches if uncorrected.

**Creating the final spindle and spectral measures.** After all filtering steps, we required at least 10 epochs (5 min) of cleaned N2 sleep for an individual to be included in downstream analyses, which removed only a handful of individuals. For EEG measures and key quantitative measures such as BMI, we set to missing outlier values defined as $\pm 3$ s.d. units from the mean. For EEG measures, we merged the two channels by averaging the individual mean values estimated for C3 and C4. Across all measures considered here, the correlations between estimates based on C3 versus C4 were typically very high. Therefore, we elected to set merged values to missing if C3 and C4 estimates were highly discrepant for a given individual. Specifically, if the difference between an individual's value for C3 and C4 was more than three times the average of the s.d.'s for C3 and C4, the merged value was set to missing. If only a single channel was available, we used the single-channel estimate as the final, merged value.

After removing outlying individuals (as described above), we were left with 3,851,924 spindles (99.8% of the original spindles retained) in 11,293 individuals (98% of individuals retained). The vast majority of individuals had $> 1$ h of filtered N2 sleep available for spindle and spectral analysis. Individuals had on average $\sim 420$ N2 epochs before any filtering. Filtering epochs based on any overlapping arousals or movements removed 113.3 epochs per individual on average. Of the remaining epochs, Buckelmüeller et al.[8] filtering removed 11.7 epochs; the second round of statistical filtering removed a further 107.6 epochs; the cardiac correction removed 23.6 epochs. The high proportion of removed epochs reflects an intentionally conservative approach rather than excessively noisy data, that is, an entire 30-s epoch will be removed even if only a 1 or 2 s are marked as containing an arousal. In the absence of outliers, iteratively applying a 2 s.d. filter three times

to an approximately normal trait would lead to $\sim 10\%$ of observations being dropped. Considering each measure individually, we indeed find that $\sim 10\%$ of otherwise unfiltered epochs are flagged; applied to eight measures (four metrics each for two channels), in total $\sim 36\%$ of otherwise unfiltered epochs are flagged for at least one measure, and so dropped from analysis. Filtered-out epochs occurred disproportionately near the beginning and end of the night (Supplementary Fig. 7).

Our conservative approach does not imply that almost half of the epochs exhibited gross artefact. To underscore this, we repeated the canonical analysis without any epoch-level filtering or correction for cardiac interference. Based on C3 (similar results obtained for C4), average spindle density was lower without filtering (1.51 versus 1.90 spindles per minute, paired $t$-test $P < 10^{-15}$, $N = 11,165$) but highly correlated (Pearson's $r = 0.92$, $P < 10^{-15}$, $N = 11,165$) with results from the filtered data set, and with broadly similar demographic associations (except correlation with BMI, arousal index and AHI were higher without epoch-filtering). The between-channel (C3/C4) correlation in spindle density was also slightly lower without filtering ($r = 0.86$ versus 0.91 for $N = 11,624$ and 11,033 respectively). This suggests that filtering enhanced spindle detection; at the same time, estimates of individual mean spindle density appeared to be relatively robust with respect to artefact, and rates of spindle activity for filtered and retained epochs were broadly similar.

All signal-processing and spindle detection steps were performed using software developed by the author (S.M.P.), as part of a C/C++ package for the large-scale analysis of sleep data (available at https://zzz.bwh.harvard.edu). All other statistical analyses were performed using R (https://www.r-project.org).

**Alternate measures of spindle activity including sigma power.** Based on fast Fourier transform and the Welch algorithm (using 4-s windows, 2-s overlap and Hanning window), we estimated spectral band power for the sigma range (defined as 12 to 15 Hz), taking the natural log of the absolute power estimate. Relative spectral power was calculated as the log transform of the absolute power per band divided by the sum of the six bands considered here: slow (0.5–1 Hz), delta (1–4 Hz), theta (4–8 Hz), alpha (8 12 Hz), sigma (12 15 Hz) and beta (15 30 Hz). The second spindle detection method, following ref. 20, applied band-pass filtering in the sigma frequency band (11–15 Hz) followed by calculating the signal root mean square in 0.25-s windows, flagging windows with values in the 95% percentile for that individual/channel. Spindles were detected and merged using a 0.3–3.0 s rule based on consecutively flagged windows, following ref. 38.

**Test–retest correlations.** For individuals with repeated polysomnography, we calculated test–retest correlations as indices of reliability, based both on raw scores and on scores adjusted for age, sex and race, using the residuals from a within-study linear regression of the measure on these predictors (in MrOS and SOF, sex was omitted). For the CHAT study, treatment arm (early adenotonsillectomy versus watchful waiting plus supportive care) did not correlate with baseline, follow-up or change in spindle density (data not shown). We additionally removed 145 individuals from the CHAT follow-up study due to inconsistent scaling of their EEG signals in the EDFs we obtained.

**Spindles and sleep macroarchitecture.** Based on manual staging from the NSRR, we calculated the duration and percentage of sleep spent in N1, N2, N3 and REM, based on all epochs (that is, irrespective of whether they were included in the spindle and spectral analyses). When estimating spindle density during N3, we applied the same pipeline as described above. Because the duration of N3 was typically shorter than N2, especially for older individuals, a greater proportion of individuals did not meet the criterion for a minimum duration of artefact-free N3 epochs. Therefore, for comparability, some analyses (for example, Supplementary Table 6) are repeated for N2 spindle density using only the subset of individuals included in the N3 analyses. Analyses of NREM 4 spindles were limited to the SHHS, which used this older staging (for other studies, all slow-wave sleep was labelled N3). Here we required at least 5 min of NREM 4 sleep, with NREM 4 epochs flanked by two other NREM 4 epochs on either side, yielding 315 individuals.

**Life-course spindle trajectories.** The association between age and spindle density is clearly nonlinear in the entire sample, thus we considered models with higher-order terms for age. Two of the six studies (those with the broadest age ranges, CFS and SHHS) showed statistical evidence for a nonlinear effect of age within that individual study (Supplementary Table 8). To simplify presentation, most analyses reported below are based on models with only linear effects: the choice of age correction did not substantively change results, as long as study membership (which is highly correlated with age) was also included as a covariate. Certain analyses (for example, the formal modelling of age effects in Fig. 2c) included higher order terms (up to the fifth), as noted in the text.

For the life-course trajectories (for example, Supplementary Figs 10 and 26), the smoothed lines were fit using the loess() function in R (with span parameter set to 1.0). The fitted curves in Fig. 2c were calculated as the expected values from five regression models of spindle density (for $F_C = 11, 12, 13, 14$ and 15 Hz) on age (including higher order terms up to the fifth), sex, study, race, BMI, arousal index

and AHI). The expectation as a function of age (from 5 to 95 years) was calculated based on the estimated intercept and five age coefficients from each regression model. Each line was scaled to have a minimum of 0 and a maximum of 1, to highlight when each trajectory peaks with respect to age.

**Frequency-dependent spindle analyses.** We repeated the wavelet analysis with different values of $F_C$ from 8 to 18 Hz in 0.25 Hz increments (referred to as the frequency-dependent spindle sets). We selected the bounds of 8 and 18 Hz to encompass the broadest range of spindle frequencies reported in the human literature. For the majority of frequency-dependent analyses, however, we focused on a more central range of frequencies, in which 11 Hz was chosen to target slow spindles and 15 Hz to target fast spindles. Importantly, our broad conclusions remained unaltered if slightly different spindle frequency values were selected from within the core range.

In these analyses, the same true spindle will typically be detected at more than one $F_C$ value. We calculated the total number of unique and discrete spindle events for each channel, by merging temporally overlapping spindles within a core frequency range ($F_C$ from 10 to 16 Hz in 0.25 Hz increments) (Supplementary Fig. 13). Specifically, we merged spindles detected on same channel, if either (a) their intersection was more than 50% of their union, or (b) more than 80% of any one spindle was overlapped by the other. The counts of merged spindles were used to estimate the total spindle density across the range of frequencies considered. These frequency-independent estimates were highly correlated with the canonical estimates ($r = 0.69$) and showed similar associations with age and sex, but had a mean of 7.4 spindles per minute, compared to 1.88. This 'total' spindle density estimate was naturally higher than the canonical estimate, which only targeted the smaller, core range of spindle frequencies, because some slower spindles were not detected with higher values of $F_C$, and vice versa.

Aggregating spindles across frequencies may not be optimal; however, it may mask important heterogeneity that is only evident when different types of spindles are analysed separately. Indeed, although estimates at nearby frequencies (for example, $F_C = 13.5$ and 14 Hz) were obligatorily highly correlated because the wavelets target overlapping ranges of frequencies, there was only very modest correlation between individuals' spindle density estimates for slow (for example, $F_C = 11$ Hz) versus fast (for example, $F_C = 15$ Hz) spindles ($r \sim 0.1$).

**Defining sleep cycles and other features of the hypnogram.** We segmented each night's sleep into cycles, following commonly adopted rules[60]. Briefly, a cycle required a minimum of 15 min of NREM sleep (with episodes starting with the onset of N2 or N3 sleep), followed by at least 5 min of REM sleep, with the exception that the first cycle allowed any duration of REM. A cycle was terminated if we observed more than 15 min of wake or N1, implying a skipped REM period for that cycle. REM periods were allowed to contain up to 15 min of NREM or wake and still be counted as a single episode. The mean number of cycles per individual was 4.4 (median 4); the mean cycle duration was 93.3 min (median, 87.3 min). For spindle analyses, we collapsed cycles 5 and above into a single category, as only 14% of individuals had more than five cycles. Supplementary Fig. 15 shows one example hypnogram and inferred sleep cycles, with ascending and descending N2 epochs[62] flagged (see below). We further defined persistent sleep as that which occurred after 10 or more minutes from previous wake. Fifty-four per cent of N2 epochs were in persistent sleep and also fell within a well-defined NREM cycle.

Epoch-level linear mixed models were performed using the lme4 R package[77], with the dependent variable set as per-epoch density (that is, two times the number of spindles observed in a 30-s epoch), separately for fast ($F_C = 15$ Hz) and slow ($F_C = 11$ Hz) spindles. Individual was included as a random effect, along with fixed effects including age (including higher-order terms up to the fifth), sex, study and race. All epoch-level analyses of within-night dynamics were based on the C3 channel only. Fixed-effect significance values were estimated from a normal approximation of the corresponding $t$-statistic. Note that epoch-level means in spindle density may differ from individual-level means, which do not weight by the number of epochs for that individual (that is, a mean across all epochs versus the mean of $N$ per-individual means).

Within-cycle variation was characterized in terms of the N2 sleep within the first six 10-min intervals of the NREM cycle, which was entered into epoch-level models as a fixed six-level factor. Descending N2 epochs transition from wake, N1 or REM sleep into N3 sleep; ascending N2 epochs are those transitioning from N3 to N1/REM/wake. For each N2 epoch, we calculated a score (ranging from $-1$ to $+1$), considering up to 5 min of preceding non-N2 epochs, coding N3 epochs as $+1$ and N1/W/R epochs as $-1$, and taking the average. A similar score was generated from up to 5 min of subsequent non-N2 sleep, but with reversed coding (that is, $-1$ for N3, $+1$ for N1/W/R). Based on the average of these two scores, N2 epochs scored above $+0.5$ were labelled as ascending, those below $-0.5$ as descending. Of 1,888,872 N2 epochs, 167,094 were labelled ascending, and 538,760 were descending.

**Chronotype and circadian effects.** Analyses of chronotype used linear regression to relate individual whole-night spindle density to sleep-midpoint controlling for

the standard covariates, allowing for both linear and nonlinear (up to fifth order) effects for sleep midpoint.

After removing individuals with an unusually short ($<30$ min) duration of N2 sleep, we fit a series of epoch-level linear mixed models, using full maximum likelihood, with per-epoch spindle density as the dependent variable. We considered only N2 epochs that occurred during persistent sleep and within a defined NREM cycle. All models had a random effect of individual, and fixed effects for age (up to fifth order), sex, study and race. Epoch-level fixed effects in all models were cycle number (1–5) and relative position within the cycle (as a six-level factor). Compared to a null model with only these terms, we fit a series of models that included one additional temporal predictor of spindle dynamics: (a) local clock time, (b) elapsed time since lights out, (c) elapsed sleep time, (d) elapsed N2 sleep, (e) elapsed N3 sleep, (f) elapsed REM sleep and (g) local clock time and elapsed sleep time. Each temporal predictor was a quantitative variable (in min) including higher order terms (up to the fifth). Clock time was coded in terms of minutes past 20:00 (with a small number of epochs occurring before this time removed from analysis). Similar results were obtained if sleep cycle number and relative position within the cycle were not included in the models. We used the Bayesian information criterion to select the best fitting model, separately for fast and slow spindles.

To infer latent circadian components, we analysed the residuals from a model predicting spindle density as a function of elapsed sleep time and the above-mentioned covariates. Specifically, we calculated their mean values in bins of epochs defined by local clock time, separately for old and young individuals. We tested whether the mean residual for each bin was significantly different from the expected value of 0 using a single-sample $t$-test. The final elapsed-sleep model used in these analyses included additional covariates to account for ascending/descending status; additionally, epoch-level fixed effects were allowed to vary as a linear function of age (similar results obtained if these additional components were not included in the final model).

**Mediation analyses for socioeconomic factors.** In testing for possible mediating effects of SES, we used the following indicators (Supplementary Table 22). In CHAT, we used parental income (variable 'par5' from sleepdata.org), parental employment status (as a binary variable, based on either mother or father indicating that they were unemployed, 'par8' and 'par9' variables) and parental education (based on the average of a rating of paternal and maternal educational attainment, variables 'par6r' and 'par7r'). For CCSHS ('ystatus' variable) and CFS ('ladder' variable), the index of SES was based on the 'ladder' self-report measure of SES (The MacArthur Scale of Subjective Social Status). For MrOS, the index of SES was based on an eight-point self-report of educational attainment ('gieduc' variable).

**CFS genetic analyses.** The CFS comprised a number of related individuals, including 369 individuals from 160 full sibships. Intraclass correlations were estimated using the ICCest() function from the ICC R package, based on the ratio of the variance of the family means to the variance of all observations from a one-way analysis of variance. Correlations are based on age- and sex-adjusted values (that is, the residuals from a regression of the measure on age and sex). For the CFS, genome-wide SNP array data were available. After imputation against the 1,000 Genomes reference panel[78], we filtered SNPs for minor allele frequency ($>5\%$), Hardy–Weinberg equilibrium ($P > 10^{-3}$), and imputation quality ($R^2 > 0.8$). We retained only autosomal SNPs, further pruning these to be in approximate linkage equilibrium using PLINK[79]. We used GCTA[69] to estimate heritabilities and genetic correlations. We excluded MZ twin pairs from the analysis (--grm-cutoff 0.75). In comparison with the intraclass correlations, which are based only on full sibships, this approach uses all individuals from different types of relationships (parent–offspring, half-sibling) as well as distantly related cousins. It also facilitates the estimation of genetic correlations (the proportion of additive genetic effects shared between two traits). Just as an observed trait variance can be partitioned into genetic and non-genetic components (that is, estimating heritability), an observed (or phenotypic) correlation can be partitioned into genetic and non-genetic components (that is, reflecting the genetic correlation). A negative genetic correlation means that genetic variants associated with increases in one trait tend to be associated with decreases in a second trait. Although GCTA is applicable to samples containing closely related individuals, there is the potential for estimates of the additive genetic variance to be confounded with shared environmental, or higher-order genetic (dominance, epistasis) variance components as is also the case for a typical sibling and twin studies. In these analyses, we therefore controlled for the major non-genetic sources of variation largely shared by siblings (age and race) as well as sex.

**Topographical spindle analyses.** A subset ($N = 53$) of the CHAT study had 18 EEG channels including frontal, central, temporal, parietal and occipital electrodes. Because topographic analyses are particularly sensitive to artefact, we took extra steps to remove spurious correlations between signals: (a) removing two individuals exhibiting uniformly low spindle density estimates across all channels, (b) removing seven individuals flagged as multivariate outliers with respect to the $2 \times 18 = 36$ ($F_C = 11/15$ Hz $\times$ 18 channels) estimates of spindle density based on the Mahalanobis distance (MVN R package), (c) removing three individuals with

excessive leverage (DFBETAS > 1) in any of the regressions of C3/C4 spindle density on spindle density at other sites and (d) regressing out the effects of age, sex and race on spindle density measures before the correlational analyses. We used the cluster (the agnes() function) and phylo packages in R to perform average-linkage agglomerative clustering using default settings, based on a matrix of distance measure calculated as $1 - |r|$, where $r$ is the correlation between spindle densities. Finally, we noted a trend whereby slow C3 spindles showed slightly greater correlations with fast spindle densities across all other channels, compared to slow C4 spindles. In contrast, fast C4 spindles showed greater correlations with fast spindles across other channels, compared to fast C3 spindles (Fig. 6). This observation (which is not driven by obvious sources of error, Supplementary Fig. 24), may point to potentially interesting laterality effects for fast and slow spindles, but requires fuller exploration in other samples.

**Data availability.** All polysomnography data are available from the National Sleep Research Resource website http://sleepdata.org. The C/C++ software used for EEG signal processing is available from http://zzz.bwh.harvard.edu.

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

## Acknowledgements

This work was supported in part through the computational resources and staff expertise provided by Scientific Computing at the Icahn School of Medicine at Mount Sinai, as well as grants from the National Institute of Mental Health (Grants MH108908 to S.M.P., MH107855 and MH099421 to D.S.M. and MH048832 to R.S.), the Stanley Center of the Broad Institute and the National Heart, Lung and Blood Institute (Grant HL114473 to S.R., supporting the National Sleep Research Resource). We also thank Leila Tarokh and also the three anonymous reviewers for helpful suggestions on the manuscript. Finally, we thank the investigators and participants who made the original studies possible.

## Author contributions

S.M.P., D.S.M., R.S. and S.R. conceived the study. All authors contributed to the analytic plan, and the interpretation of results. S.M.P. developed the analytic tools used in this study and performed the primary analyses, with support from C.D., B.E.C., S.M., R.C. and G.P. S.M.P. drafted the initial manuscript; all authors reviewed and revised the final manuscript.

## Additional information

**Competing interests:** The authors declare no competing financial interests.

