## [Peer Review File · Nature Communications]

Reviewers' comments:

Reviewer #1 (Remarks to the Author):

This manuscript by Purcell and colleagues is a colossal work that enlightens the phenomenon of sleep spindles from a number of different angles. It is unique in terms of data base, analysis methods, and the breadth of topics it covers. It reproduces some known facts about sleep spindles in a huge sample size, it reports a number of novel observations e.g. on the heritability of sleep spindle characteristics, and it discusses methodological issues regarding sleep spindle detection. Particular achievements of this manuscript are providing several methodological approaches to the investigation of the genetic basis of spindles, and the discussion of the relation between spindle amplitude and spindle density. It is especially commendable that the authors have actually used and compared two algorithms for sleep spindle detection. This is an excellent piece of work that, however, can be improved mainly by making it more concise.

Blaise Pascale is quoted having written in a letter: "I have made this longer than usual because I have not had time to make it shorter." This manuscript (133 pages, 43 tables and 40 figures) would also greatly benefit if the authors would spend some more time on sharpening its messages. My fear is that the most important sections of the manuscript, which can be found in the latter part of the manuscript, will not receive the attention they deserve because the readers will capitulate before reaching them. I believe that the manuscript will receive more attention if the length (incl. supplement) could be cut in half. There are a number of topics, figures and tables that are of lesser importance to the paper and could be removed. Some suggestions, which the authors are free to ignore: Tables S2, S7, S8 (laterality effects), S11, S12, S15, S19, S20, S21, S24, S25, S29. Figures 2, 7, 8, S3, S8, S12, S14, S18, S19, S21, S24, S25, S27, S30.

Currently I had to read both documents (MS and SI) open in parallel to understand methods, results and discussion of each topic. I would prefer to have the more important topics entirely in the main manuscript and the less important ones in the supplement. It might also be considered to provide a distinct methods section.

All analyses relating to age suffer from the confound that age is mainly a between-study factor. This means that different age groups were recorded in different environments, with different recording equipment, and probably with slight differences in electrode positioning. These differences (e.g. the distance between recording electrode and reference, or the recording impedance) can affect signal amplitude. Three possible solutions come to my mind: 1) add within-study analyses of the age effect, 2) reduce the relevance of age effects in the manuscript, and/or 3) discuss this problem in more detail.

A central point of the paper is that spindle amplitude can explain observed spindle density to a large extent. This explanation might also be applied to other observations throughout the manuscript. Race effects might be influenced by anatomical differences in skull thickness. Ageing is known to change skull thickness, and it shrinks the brain. Males and

females also have differences in skull thickness. As distance of the brain surface to the recording electrode affects EEG amplitude, these factors should be discussed. In the same vein, it has also been discussed that age-related decreases in SWS are mainly related to lower EEG amplitudes no longer reaching the criterion for SWS. This discussion might also be referenced.

The analysis on 'ascending' and 'descending' sleep spindles is probably confounded with circadian time and/or previous time asleep. This analysis should be performed within each 90-min sleep cycle. As the analysis does not add much to the main messages of this paper, it might also be entirely removed (and perhaps reported in more detail elsewhere). Again, this is just a suggestion.

While I agree that age and sex might well be related with spindle amplitude rather than spindle density, I believe benzodiazepines might not. The type and number of spindles induced by benzodiazepines are – at least visually – quite conspicuous. There might actually be some references to that, which I sadly do not have at hand at the moment.

I fully agree that trait-like genetic influences on sleep spindles and those related to age, sex and race might well manifest themselves in spindle amplitude rather than density. On the other hand, learning induced changes in spindle density, which occur in a state-like fashion, might on the other hand actually increase spindle density. Although this is just speculation, it might be worth mentioning this in the discussion.

The authors use slow and fast spindle frequencies in their analyses throughout the manuscript. However, they only briefly discuss whether these two types of spindles actually exist. Other options would be that there is only one type of spindle which changes its frequency throughout the night, or that there is a continuous spectrum of spindles with any frequency within a certain range. This discussion might be extended.

Figure 5c seems somehow misleading because a normalized spindle density is given. It would be more informative with absolute densities.

Figs. 7 and S25 are not very useful because normalization produces a plot that is largely dependent on the ratio of black and white patients in the sample. It might be more useful to show simple difference spectra between white and black patients instead. Or normalization should be done in a way that group values are not dependent. Altogether, these figures provide a lot of confusing data, but only little insight, and might therefore be removed.

Tables S6 and S14 convey somehow a contradictory message. Retesting shows highly significant correlations but also highly significant differences for the same measures. Do spindles belong to an individual 'EEG fingerprint' or do they change over time?

The problem with ECG contamination of the EEG and the relation to BMI is somehow a sideline in the manuscript, which might be moved entirely to the supplement.

In Figure S7 it seems that the ECG artefact cannot be removed entirely. This might still have affected the T1-3/TA-analyses. Maybe all channels showing ECG artefacts should be removed? This might be particularly relevant for the genetic correlations.

Figure S18 is particularly confusing because it depicts the opposite of what the analysis shows.

The argument in Figure S30 seems flawed. A larger spindle amplitude would also engender new spindles with very small amplitudes. Increasing amplitude would therefore in my opinion result in a broader distribution with a peak shifted to the right.

Effect sizes are given only in a few places. It would be desirable to have them throughout the manuscript.

Some analyses have larger numbers of tests. Were corrections for multiple testing applied?

Why are so many epochs (~45%) removed from analysis?

For many analysis, the reader does not exactly know which factors and which covariates entered into analysis. This should be checked and given for each analysis in a standardized way.

Similarly, the tables seem sometimes to have different orientations, i.e. predictors and dependent variables swap positions. This could confuse the reader.

The use of the word 'phenotype' seems unusual. I only know it to mean the actual ensemble of observable characteristics displayed by an organism (e.g. a 'fast spindle' and a 'slow spindle' phenotype), but not as a synonym for 'characteristic' or 'trait' (e.g. spindle frequency).

When first mentioning the band-pass analysis on page 5, mention already that results are reported only in the end.

Page 11, first sentence: were the drug_s_

Page 17: "careful matching on race race"

Figure 10: Add a colour scale.

Table S29: Are the values for 'household income' correct?

Table S35: Describe T1-3, TA and Default in table legend.

Table S35: "Spindle density estimates based on lower detection thresholds tend to be less reliable, as indexed by the test/retest correlation." Does this fit with the data (T1>T3>T2)?

Table S39: Explain the joint models.

Figure S14: What is correlated?

Figure S15: (8 to 18 Hz)

Figure S17: What is on the x-axis?

Figure S22: Y-axis is number of patients?

Figure S26: Provide colour scales.

Reviewer #2 (Remarks to the Author):

Summary:

The authors used data from several large studies in which EEG was collected to study the properties of sleep spindles and the effects of age, sex, race and other factors. Their analyses of this very large dataset provide a valuable contribution.

Comments:

Page 4, paragraph 1, line 5: I would say "evaluated" instead of "ascertained"

Page 5, first paragraph, line 4: "cardiac interference in the EEG" would be clearer than "of the EEG"

Page 6, paragraph 3: Are the statistically significant differences observed between hemispheres consistent across individuals, and is any variation between individuals correlated with dominant handedness?

Page 7, paragraph 1: I would say "nonlinear" rather than "non-linear"

Page 11, paragraph 2, line 1: I would say "evaluated for" or "diagnosed with", as appropriate, rather than "ascertained for"

Page 11, paragraph 3: I find it somewhat surprising that the spindle occurrences in N2 and N3 sleep are this close. This might deserve more comment, since spindles are so often associated so strongly with N2 sleep in particular. I would also be interested to see differences between N3 and N4 sleep if any of the available data was scored according to the older stages.

Page 16, paragraph 2, line 4: uV should be replaced with μV

Page 16, paragraph 2: Table S35 includes T_A in addition to the thresholds described here. What is T_A? It also seems interesting/noteworthy that the T_2 threshold results in so few detections relative to the lower and higher thresholds.

Page 18, paragraph 5: Since, as the authors point out, human scoring of spindles tends to be highly inconsistent, I might argue that automated scoring is closer to optimal.

TABLES AND FIGURES:

Table 2: I would be interested to these results for N3 and N4 (according to pre-2008 scoring standards) separately, if there is any such data available, since there are real differences between the stages.

In general, I think it would be helpful to consistently provide units with all of the measures in all the figures, despite many of them being the same throughout.

Figure 3: It seems like the lowest bin for the spindle duration histogram is for durations between 0.5 and 0.6 s, so it seems somewhat confusing start the x-axis at 0.6. Also, this figure is nice, but I would like to see some of this general information on the properties of spindles across the population tabulated too.

Figure 4: I would move the x-axis labels to below the female plots.

Figure 7: It's stated in the caption, but I think a legend for the color of the lines corresponding to the two racial groups would also be helpful.

Figure 8: I would move the sentence "Note, because of the normalization, the profile..." to before the sentence "The third column shows,..." in this caption. Currently it seems to be referring to the plots in the third column, although I don't think that is the case.

Figure 9: The asterisk on the non-significant p-value for the SOF comparison is not actually defined.

Figure 10: There should be a color scale for the r_g plots. There's also an extra hyphen at the end of the second line of the caption.

Reviewer #3 (Remarks to the Author):

This is weighty article in which a huge effort has been made for describing, for providing normative values, and for defining "the state of art" of our knowledge on sleep spindles (SS). The other face of this article is that majority of findings are not original.

The evaluation of a sleep researcher qualifies it as a probable milestone in studies on SS.

On the whole, I have found it intellectually honest.

As known by the authors, the principle limit of the study is represented by the lack of any regional (topographical) information. In fact, most of well described phenomena of the current study interact or covary with regional changes. This is not merely a missing finding.

Another important missing issue regards the circadian modulation of SS (e.g., Knoblauch et al. *Sleep*. 2005; 28:1093-101; Knoblauch et al., *Eur J Neurosci*. 2003;18:155-63; Wei et al., *Neurosci Lett*. 1999; 260: 29-32). At variance of topography, I suppose that the authors can make an effort in this direction, and they have to select data collected in different times of day/night.

Moving to the specific points:

1. In my opinion, the analyses on the spindle density and sleep stage duration could be drastically reduced due to the fact that they are mostly obvious;
2. Data on inter-hemispheric EEG (C3, C4) spectral coherence strictly resemble those on raw EEG power values (Figure S11). This deserves a comment.
3. The lack of evidence for a bimodal distribution of spindle frequency seems misleading. It seems a by-product of: (A) the lack of frontal recordings (i.e., the "lacking" second peak is maximal at these leads), (B) the fact that the "canonical" analysis is centered on 13.5 Hz;
4. The large difference between "frequency-independent" and "frequency-dependent" estimates (7.4 vs. 1.88) should be better clarified
5. When describing life course trajectories of frequency-dependent spindle density, the authors conclude that "it will be challenging to interpret changes in spindle activity if age and spindle frequency are not taken into account". This is true, but even more is challenging if age and spindle frequency do not take into account regional changes (e.g. for infants: Novelli et al., *J Sleep Res*. 2016; 25: 381-9; for elderly: Martin et al., *Neurobiol Aging*. 2013; 34: 468-76). Please, discuss in some way also this issue
6. Analyses regarding spindle trajectory over the course of the night are suboptimal. Please, plot changes of spindles across consecutive sleep cycles/periods. Sleep periods more than time spent in N2 sleep describes the time course of SS across the night. Furthermore, this approach will make results more comparable to those previously published
7. Please, de-emphasize relevance of genetic results. They do not add so much to the existing knowledge (e.g., Ambrosius et al., *Biol Psychiatry*. 2008; 64: 344-8; De Gennaro et al., *Neuroimage*. 2005; 26: 114-22; De Gennaro et al., *Ann Neurol*. 2008; 64: 455-60; Adamczyk et al., *Front Hum Neurosci*. 2011; 9: 624)

Response to Reviews

We thank all three Reviewers for their helpful comments and their suggested changes and additions: we have addressed the issues raised and feel that the manuscript is significantly improved.

Before addressing comments point-by-point, we first summarize the five most important additions/changes made by the Reviewers and Editors.

1) Analysis by NREM sleep cycle

We implemented a standard heuristic (following Feinberg & Floyd, 1979) to define NREM sleep cycles, applying it to all NSRR records. We now present the primary description of spindle dynamics across the night in terms of sleep cycles. Although our previous observations about spindle dynamics still hold, we agree that this mode of analysis and presentation is superior.

As well as considering changes in spindle activity *between* sleep cycles, we also considered variation *within* each sleep cycle, in terms of elapsed time (i.e. **Figure 3a**) but also in terms of “local context” – the relative position within the hypnogram. Specifically, we considered spindles in a) ascending versus descending N2 epochs (as previously reported), b) at NREM/REM transitions, c) near sleep onset versus during persistent sleep. In a series of analyses that control for sleep cycle, we show the relevance of these ultradian factors and highlight examples of fast and slow spindles exhibiting functionally distinct associations. Our previous observations regarding spindles during descending versus ascending N2 still hold, even after controlling for sleep cycle and other factors (**Figure 3b**).

New material: as well as sections in the main text, relevant Tables/Figures: Figures 3, S15 - S17; Table S12

2) Circadian modulation of spindle activity

In addition to the ultradian effects above (i.e. sleep cycles, ascending/descending N2, NREM/REM transitions, etc), we report a new analysis of circadian influences on spindle dynamics. Naturally, to unambiguously disentangle circadian from sleep homeostatic and other ultradian factors requires forced desynchrony or constant routine experimental paradigms. Nonetheless, we believe there is value in a sufficiently rigorous “big data” analogue, particularly in the context of deriving phenotypes for large-scale genetic studies (see below).

We first considered individual-level chronotype, as indexed by sleep midpoint (not finding any robust associations with whole-night spindle parameters).

We next revised our epoch-level analyses to model putative circadian influences. Specifically, we fit a series of linear mixed models in which spindle density varied as a function of a) local clock time, b) elapsed time since lights out, c) elapsed sleep, d) elapsed N2, e) elapsed N3 or f) elapsed REM. All models allowed for nonlinear, higher-order features and controlled for individual and ultradian factors. Unlike the elapsed time measures, which by definition cannot capture any of the between-individual variation that is due to circadian factors, clock time (indirectly and approximately) indexes circadian phase as well as sleep homeostatic and ultradian effects.

Based on the Bayesian information criterion, which conservatively accounts for model complexity, the “elapsed sleep” model fit the data best for both fast and slow spindles (**Figure S18**). The fit was further improved by adding clock time, suggesting that circadian factors (i.e. as approximated by variation in clock time that is independent of variation in elapsed time) have an impact on spindle activity over and above sleep homeostatic and ultradian factors, as the previous literature has shown.

We then attempted to characterize the circadian modulation of fast and slow spindles in the NSRR.

Previous reports (e.g. Dijk & Czeisler, 1995; Knoblauch et al., 2005) have noted a) that fast and slow spindles have 180-degree phase-shifted rhythms (slow spindles peak around the time of maximum melatonin levels, whereas fast spindles have their nadir, i.e. typically around 3-4am), and b) that circadian modulation of spindles is attenuated in older individuals.

Critically, we assumed that after accounting for the effects of elapsed sleep on spindle activity, any residual covariation with local clock time represented only circadian modulation. As shown in **Figure S19**, which plots the residuals from the elapsed sleep model against clock time, we were in fact able to recapitulate the two main features highlighted above: qualitatively different circadian modulation of fast and slow spindles that was present only in younger individuals.

While not a gold-standard analysis of circadian modulation, this approach points to the potential for “scoring” large samples of individuals in terms of the magnitude or type of their circadian modulation. For example, one could use the cross-covariance with elapsed-sleep model residuals against a known or inferred circadian signature to reflect the extent of circadian modulation in a given individual for fast or slow spindles. Instead of comparing young versus old individuals, one can imagine comparisons by genotype, or by disease state, etc. We briefly propose this idea in the revised manuscript and discuss potentials and limitations.

New material: as well as sections of the main text, relevant Tables/Figures: Figures S18, S19.

3) Topographical analyses

To address the relative paucity of topographical information in most NSRR studies, we’ve included a new analysis that focuses on a subset of the CHAT study ($N=53$) for whom EEG data were collected from 18 scalp electrodes.

As well as replicating expected topographical effects (i.e. that slower spindles are predominantly observed at frontal electrodes), our main focus was to estimate the extent to which future (genetic) studies of spindles based on a limited EEG montage (e.g. only C3) will be able to capture individual differences more globally (**Figure 6**). Although slow spindles were in fact more prominent at frontal scalp electrodes, the majority of the inter-individual variation in “slow, frontal” spindles was actually captured by slow spindle density estimated at C3 (**Figures 6, S24, S25**).

More nuanced topographical questions (e.g. concerning connectivity, relative topographic profiles or spindle propagation) will naturally be impossible in a study with only one or two central electrodes. Nonetheless, this new analysis makes the important points that a) most of the between-individual variability in within-channel spindle density relates to frequency rather than topography, and consequently, b) that in future (genetic) studies analyzing fast and slow spindles separately will be more important than analyzing frontal and centroparietal spindles separately.

New material: as well as sections of the main text, relevant Tables/Figures: Figures 6, S24, S25.

4) Spindles in NREM 4 in sleep

As requested, we determined that some polysomnograms (SHHS) had information from the older, Rechtschaffen and Kales staging system that differentiated N3 into NREM 3 and NREM 4 stages. Extending the previous section that contrasted N3 and N2 spindles, we indeed detected spindles during NREM 4 sleep. NREM 4 spindles were qualitatively similar to NREM 2 and 3 spindles in terms of demographic associations, albeit occurring significantly less frequently. Importantly, correlations in spindle density from NREM 2, 3 and 4 were all high. We argue that these results support our previous conclusion, namely that spindle activity persists throughout the NREM cycle.

New material: as well as sections of the main text, relevant Tables/Figures: Figures S14.

5) Eliminating unnecessary Tables/Figures

Reviewer 1's comments on the length and structure of the manuscript were well taken.

We've restructured the main text (now with an explicit Methods section) to be shorter and more self-contained. With the exception of some brief text describing each study, the Supplement now contains only a series of supplementary Tables and Figures and the corresponding legends, rather than the extensive methodological notes of the original version.

Even after adding the new material requested by the Reviewers and Editors, there are in total 14 fewer Tables and Figures (and there are now fewer than 10 main text Figures/Tables, consistent with *Nature Communications* publication policies).

SPECIFIC COMMENTS

Below we address the reviewer's comments point-by-point. Key points within longer comments are **bolded** and our responses are in **blue type**.

Reviewer #1 (Remarks to the Author):

This manuscript by Purcell and colleagues is a colossal work that enlightens the phenomenon of sleep spindles from a number of different angles. It is unique in terms of data base, analysis methods, and the breadth of topics it covers. It reproduces some known facts about sleep spindles in a huge sample size, it reports a number of novel observations e.g. on the heritability of sleep spindle characteristics, and it discusses methodological issues regarding sleep spindle detection. Particular achievements of this manuscript are providing several methodological approaches to the investigation of the genetic basis of spindles, and the discussion of the relation between spindle amplitude and spindle density. It is especially commendable that the authors have actually used and compared two algorithms for sleep spindle detection. **This is an excellent piece of work that, however, can be improved mainly by making it more concise.**

Blaise Pascale is quoted having written in a letter: "I have made this longer than usual because I have not had time to make it shorter." This manuscript (133 pages, 43 tables and 40 figures) would also greatly benefit if the authors would spend some more time on sharpening its messages. My fear is that the most important sections of the manuscript, which can be found in the latter part of the manuscript, will not receive the attention they deserve because the readers will capitulate before reaching them. I believe that the manuscript will receive more attention if the length (incl. supplement) could be cut in half. There are a number of topics, figures and tables that are of lesser importance to the paper and could be removed. Some suggestions, which the authors are free to ignore: Tables S2, S7, S8 (laterality effects), S11, S12, S15, S19, S20, S21, S24, S25, S29. Figures 2, 7, 8, S3, S8, S12, S14, S18, S19, S21, S24, S25, S27, S30.

Currently I had to read both documents (MS and SI) open in parallel to understand methods, results and discussion of each topic. I would prefer to have the more important topics entirely in the main manuscript and the less important ones in the supplement. It might also be considered to provide a distinct methods section.

Agreed. As noted above, we have reduced the total number of tables and figures (14 fewer in total, even after adding new material as requested by the Reviewers and Editors). Methodological details in the Supplement have been moved to the Methods section of the main text, to make the main document more "standalone". Whilst still a lengthy manuscript, we hope this is justified by the scope of topics covered.

All analyses relating to age suffer **from the confound that age is mainly a between-study factor**. This means that different age groups were recorded in different environments, with different recording equipment, and probably with slight differences in electrode positioning. These differences (e.g. the distance between recording electrode and reference, or the recording impedance) can affect signal amplitude. Three possible solutions come to my mind: 1) add within-study analyses of the age

effect, 2) reduce the relevance of age effects in the manuscript, and/or 3) discuss this problem in more detail.

The original manuscript in fact analyzed age effects in three ways, specifically to address this concern. Within-study analysis (old Table S12, now S8) and a longitudinal within-individual analysis (old Table S14, now S9) produced results consistent with the primary analysis in the entire sample (which itself controlled statistically for study as a covariate). The reduced length of the manuscript should help to avoid these qualifying analyses being overlooked.

A central point of the paper is that spindle amplitude can explain observed spindle density to a large extent. This explanation might also be applied to other observations throughout the manuscript. **Race effects might be influenced by anatomical differences in skull thickness. Ageing is known to change skull thickness, and it shrinks the brain. Males and females also have differences in skull thickness.** As distance of the brain surface to the recording electrode affects EEG amplitude, these factors should be discussed. In the same vein, it has also been discussed that age-related decreases in SWS are mainly related to lower EEG amplitudes no longer reaching the criterion for SWS. This discussion might also be referenced.

We have added text noting these possibilities. Although other studies have concluded that skull thickness is a negligible determinant of individual differences in scalp EEG (Hagemann et al., 2008), we note these possibilities, particularly in relation to the observed race effect. We note that it is still important to document, if not explain, such effects inasmuch as disease biomarker studies that measure spindle activity still risk confounding, if factors such as age, sex and race are not taken into account (i.e. whether or not the differences are “real” or epiphenomenal, it could still lead to false conclusions about the functional roles of spindles).

The analysis on ‘ascending’ and ‘descending’ sleep spindles is probably confounded with circadian time and/or previous time asleep. **This analysis should be performed within each 90-min sleep cycle.** As the analysis does not add much to the main messages of this paper, it might also be entirely removed (and perhaps reported in more detail elsewhere). Again, this is just a suggestion.

As noted above, we have revised the analysis of spindle dynamics to focus primarily on sleep cycles (i.e. **Figure 3b**). As well as documenting between-cycle effects, we also consider within-cycle effects including the distinction between ascending and descending N2 sleep and other ultradian features. Our previous conclusions on spindle activity in ascending versus descending N2 sleep hold. (Note: the original analysis of ascending/descending sleep did in fact control for elapsed sleep, although we agree that conditioning on sleep cycle is a preferable approach.)

While I agree that **age and sex might well be related with spindle amplitude rather than spindle density, I believe benzodiazepines might not.** The type and number of spindles induced by benzodiazepines are – at least visually – quite conspicuous. There might actually be some references to that, which I sadly do not have at hand at the moment.

We agree that different mechanisms could be at play and we leave it as an open, empirical question – our intention was to highlight the need to consider these different aspects of spindle activity and the extent to which measurement issues confound them. Along these lines, we point to the text in the Discussion: *“Whether or not spindle density and amplitude are truly correlated across individuals (as opposed to this correlation being artificially induced by the choice of amplitude threshold) is also an open and empirical question. For example, benzodiazepine use was still associated with (independent) increases in both spindle density and amplitude under the T_A set (data not shown).”*

I fully agree that trait-like genetic influences on sleep spindles and those related to age, sex and race might well manifest themselves in spindle amplitude rather than density. **On the other hand, learning induced changes in spindle density, which occur in a state-like fashion, might on the other hand actually increase spindle density.** Although this is just speculation, it might be worth mentioning this in the discussion.

Agreed, we have added text to allude to this possibility in the Discussion.

The authors use slow and fast spindle frequencies in their analyses throughout the manuscript. **However, they only briefly**

discuss whether these two types of spindles actually exist. Other options would be that there is only one type of spindle which changes its frequency throughout the night, or that there is a continuous spectrum of spindles with any frequency within a certain range. This discussion might be extended.

Agreed, this is an interesting point, and we have added some text to the Discussion along these lines. In our opinion, the functional dissociations that we (and others) have presented provide the most compelling evidence for different types of spindles. (Between individuals, if there were only one type of spindle that varied in its typical frequency, we'd presumably expect a *negative* correlation in fast versus slow spindle density, which is not what we see, i.e. as people with "faster spindles" would necessarily have more "fast spindles" but also fewer "slow spindles"). Whether or not there are precisely two truly discrete types, what the associated frequencies ranges are, and to what extent those ranges vary between individuals, remain open questions.

Figure 5c seems somehow misleading because a **normalized spindle density** is given. It would be more informative with absolute densities.

Our aim was to emphasize the developmental trend by which faster spindles peak at later ages. We have elected to keep this representation, which places greater visual emphasis on the developmental change within each spindle definition. We note that information about absolute density as a function of age is presented in the other two panels of the same Figure. Both versions of the figure are shown below: we'd argue that neither presentation is "wrong", but the relative version more clearly illustrates our point.

Figs. 7 and S25 are not very useful because normalization produces a plot that is largely dependent on the ratio of black and white patients in the sample. It might be more useful to show simple difference spectra between white and black patients instead. Or normalization should be done in a way that group values are not dependent. Altogether, these figures provide a lot of confusing data, but only little insight, and might therefore be removed.

Agreed, the normalization was visually confusing. We have changed this Figure: we now present the spindle densities for blacks only, after normalizing by the mean and standard deviation for whites. That is, the plot shows how the mean of the black group varies in terms of the range of variation within the white group, which is no longer dependent on relative sample sizes, and has a more straightforward interpretation (i.e. SD unit differences). (Note, old Figure 7 – which was a subset of old Figure S25 – has been removed; old Figure S25 is now S21.)

Tables S6 and S14 convey somehow a contradictory message. Retesting shows highly significant correlations but also highly significant differences for the same measures. Do spindles belong to an individual 'EEG fingerprint' or do they change

over time?

We disagree that there is any inherent contradiction here, and have made the text clearer and more explicit about this. Age-dependent mean shifts can be consistent with substantial and stable individual differences. An analogy with height, a highly heritable trait, is useful here: that height changes both dramatically and predictably over time (i.e. children get taller) is not at odds with stable effects (from genes and other factors) that persist over time. That is, compared to their peers, taller children will typically grow to be taller adults. Similarly, individuals follow genetically influenced, predictable trajectories in how spindle activity changes over the life-course.

The problem with ECG contamination of the EEG and the relation to BMI is somehow a sideline in the manuscript, which might be moved entirely to the supplement.

Although this observation is very much in line with the aims of our manuscript (i.e. to characterize sources of variation in measured spindle activity so as to inform robust biomarker and genetic studies), we agree that this observation can be well dealt with in the Methods/Supplement. We have moved this Figure to the Supplement and now only mention this issue briefly in the main text (with details in the Methods section).

In Figure S7 it seems that **the ECG artefact cannot be removed entirely**. This might still have affected the T1-3/TA-analyses. Maybe all channels showing ECG artefacts should be removed? This might be particularly relevant for the genetic correlations.

The issue here is that the cardiac/ECG artefact is not an all-or-nothing phenomenon. The original correlation between spindle density and BMI was not driven only by a handful of clearly aberrant individuals, as we note in the Supplement/Methods. As such, a strategy of removing only the most egregious cases of artefact will not work well (or will remove too many channels). We agree that future approaches for removing this artifact may improve on what we've presented here, and that particular types of spindle analyses may be more sensitive to any residual artefact.

Figure S18 is particularly confusing because it depicts the opposite of what the analysis shows.

Agreed. We dropped this Figure, as age effects are more clearly demonstrated elsewhere in the manuscript. (Although using the loess curve provided a simple, intuitive way to describe age effects, as we noted in the legend, it failed to capture the true inverted-U trajectory of slower spindles evident in the formal statistical analyses.)

The **argument in Figure S30 seems flawed**. A larger spindle amplitude would also engender new spindles with very small amplitudes. Increasing amplitude would therefore in my opinion result in a broader distribution with a peak shifted to the right.

We are not sure how to interpret this. Conceptually, it is not obvious that increased amplitude should necessarily increase true spindle density (i.e. "engender new spindles") – we are not aware of any biological or theoretical justifications for this assumption. We present one simple and parsimonious model in which true density and amplitude are independent, making the point about differential detectability, although we agree that other true models are in principle possible. Along these lines – we do note in the text that the true relationships between different spindle properties are not well characterized, but an important first step to develop methods that can better isolate them.

Effect sizes are given only in a few places. It would be desirable to have them throughout the manuscript.

A central aim of the manuscript is to characterize the heterogeneity of spindle activity. Heterogeneity means that a single parameter often is not sufficient to describe the whole population. This fact is not unrelated to the relatively large number of Supplementary Tables and Figures, which contain effect size estimates (e.g. regression coefficients or group means), but that are often stratified by age, sex, and other individual-level and spindle-level factors that index heterogeneity. Relating this information

directly in the main text is likely to be unwieldy or misleading.

Some analyses have larger numbers of tests. Were corrections for **multiple testing** applied?

We do present multiple analyses to address multiple, inter-related hypotheses, although the number is small by many contexts, e.g. genome-wide association studies testing millions of variants in a fixed and well-defined framework. Unlike GWAS, the number of independent hypotheses and tests is not well defined here, so it is neither straightforward nor necessarily desirable, to apply an “experiment-wide” level of correction. In almost all cases however, the large sample size means that key results are likely to withstand any correction method, i.e. many results are reported as $p < 10^{-15}$ – under Bonferroni correction for an experiment-wide type I error rate of 5%, this allows for one quadrillion (one thousand million million) independent tests. For almost all results presented, the strength of the statistical evidence is several orders-of-magnitude greater than previous reports on spindles have offered.

We ensure statistical rigor by a) transparently presenting virtually all tests performed (i.e. in the numerous Supplementary Tables) and b) often looking for consistency within the six NSRR studies as an internal replication, as well as with the existing literature where appropriate.

Why are so **many epochs (~45%) removed** from analysis?

We have added text to the Methods section (“Creating the final spindle and spectral measures”) to describe this. In short, we chose to aggressively filter the data because we could do so and still retain more than enough data to infer spindle activity. (We wanted a pipeline that maximized the number of individuals, as will be important for genetic studies, rather than the amount of signal per individual. As such, the pipeline is conservative in terms of which epochs are retained.)

Importantly, our conservative approach does not imply that almost half the epochs exhibited gross artifact or were “bad”. To underscore this point, we have added an analysis in which absolutely no filtering was applied, which actually yields similar (albeit more noisy) estimates of spindle density. See the abovementioned section for more details.

For many analyses, the reader does not exactly **know which factors and which covariates entered into analysis**. This should be checked and given for each analysis in a standardized way.

We have added text in the Methods section that describes the standard sets of covariates applied to all analyses unless otherwise noted.

Similarly, **the tables seem sometimes to have different orientations**, i.e. predictors and dependent variables swap positions. This could confuse the reader.

We agree this is not ideal, but because of the various stratifications by study or other factors, some tables are intrinsically different from others, meaning a one-size-fits-all layout is not ideal either, as it can lead to problems with pagination or readability.

The use of the word ‘phenotype’ seems unusual. I only know it to mean the actual ensemble of observable characteristics displayed by an organism (e.g. a ‘fast spindle’ and a ‘slow spindle’ phenotype), but not as a synonym for ‘characteristic’ or ‘trait’ (e.g. spindle frequency).

Agreed, we have replaced phenotype with ‘trait’, ‘character’ or similar. (Although it is a standard in the genetics literature to refer to an isolated quantitative trait or disease as a ‘phenotype’, we agree that this is not primarily a genetics manuscript and so we need not use the jargon from that field.)

When first mentioning the **band-pass analysis** on page 5, mention already that results are reported only in the end.

In shortening the manuscript, we now only introduce the band-pass analysis later in the text, reporting

the results at the same time.

Page 11, first sentence: were the drug_s_

Corrected.

Page 17: "careful matching on race race"

Corrected.

Figure 10: Add a colour scale.

Corrected.

Table S29: Are the values for 'household income' correct?

Corrected: group means for blacks and whites were indeed swapped. Also, we now flag more clearly that the income variable is based on an 8-point scale (rather than \$ amount). We've also updated presentation to more clearly reflect the direction of effects for race, making it consistent with other tables (i.e. beta represents the effect associated with being black).

Table S35: Describe T1-3, TA and Default in table legend.

Done (is now Table S30). Note, to simplify analysis, we've omitted T_2 and T_3 groups as they didn't strengthen or alter the conclusions, but they did cause unnecessary confusion.

Table S35: "Spindle density estimates based on lower detection thresholds tend to be less reliable, as indexed by the test/retest correlation." Does this fit with the data ($T_1 > T_3 > T_2$)?

As above, T_2 and T_3 have been removed from the manuscript. (The different threshold intervals used obligatorily implied different numbers of spindles, meaning differential measurement error, etc. These trivial factors distracted from the primary point, i.e. that there is considerable meaningful sub-threshold spindle activity.)

Table S39: Explain the joint models.

Text added to the legend (now Table S33).

Figure S14: What is correlated?

Individual's spindle density. (Note: in the interest of shortening the manuscript, this Figure has been removed.)

Figure S15: (8 to _1_8 Hz)

Corrected.

Figure S17: What is on the x-axis?

Spindle density. (Note: in the interest of shortening the manuscript, this Figure has been removed.)

Figure S22: Y-axis is number of patients?

Number of epochs. (Note: in the interest of shortening the manuscript, this Figure has been removed.)

Figure S26: Provide colour scales.

Note: in the interest of shortening the manuscript, this Figure has been removed.

Reviewer #2 (Remarks to the Author):

Summary:

The authors used data from several large studies in which EEG was collected to study the properties of sleep spindles and the effects of age, sex, race and other factors. Their analyses of this very large dataset provide a valuable contribution.

Comments:

Page 4, paragraph 1, line 5: I would say “evaluated” instead of “ascertained”

We appreciate the ambiguity of the word “ascertained”. Here we referred explicitly to study ascertainment: i.e. a proportion of individuals were specifically recruited into the study because of their sleep apnea symptoms. Because “evaluated” would change the meaning of the sentence, we’ve replaced this term with “recruited”.

Page 5, first paragraph, line 4: “cardiac interference in the EEG” would be clearer than “of the EEG”

Agreed, corrected.

Page 6, paragraph 3: Are the statistically significant differences observed between hemispheres consistent across individuals, and is any variation between individuals correlated with dominant handedness?

The mean difference between spindle density at C3 versus C4 is statistically significant but modest. If “consistent across individuals” means “do all individuals show this laterality?”, then answer is no. Only 52% of the sample have C3 > C4 (although this is highly significantly different from 50%). If “consistent across individuals” means “does an individual’s degree of laterality persist over time?”, then the answer is yes. Looking at test / retest correlations for the difference between C3 and C4 density (either in absolute terms, or expressed as a proportion of their mean spindle density across both C3 and C4), we see highly significant test / retest correlations in CHAT, SHHS and MrOS. As for handedness, we only had access to this information for the CHAT study, in which 45 children were reported to be left-handed. There was no association between handedness and this measure of “spindle laterality” (either controlling or not for mean spindle density and other covariates such as age, sex, etc).

Page 7, paragraph 1: I would say “nonlinear” rather than “non-linear”

Agreed, corrected.

Page 11, paragraph 2, line 1: I would say “evaluated for” or “diagnosed with”, as appropriate, rather than “ascertained for”

See above.

Page 11, paragraph 3: I find it somewhat surprising that the spindle occurrences in N2 and N3 sleep are this close. This might deserve more comment, since spindles are so often associated so strongly with N2 sleep in particular. I would also be interested to see differences between N3 and N4 sleep if any of the available data was scored according to the older stages.

As described above, we were able to find a subset of the NSRR for which N4 sleep was discriminated from N3 sleep. We have included this new analysis, which finds a lower but detectable rate of

spindles, which show similar properties to N2 spindles in key regards.

Page 16, paragraph 2, line 4: uV should be replaced with μV

Agreed, corrected.

Page 16, paragraph 2: Table S35 includes T_A in addition to the thresholds described here. What is T_A? It also seems interesting/noteworthy that the T_2 threshold results in so few detections relative to the lower and higher thresholds.

T_A is now better described; also, as noted above, in the interest of simplicity we have removed T_2 and T_3 thresholds. (The number of detections in a given interval is only a trivial function of the size of the interval, i.e. an interval set at 4 to 4.0001 would probably not detect any spindles. For this reason, it was clearer to remove these additional intervals; this does not substantively change any results or conclusions.)

Page 18, paragraph 5: Since, as the authors point out, human scoring of spindles tends to be highly inconsistent, I might argue that automated scoring is closer to optimal.

We agree (and think that the results presented for low-amplitude T_1 spindles and also for N4 spindles support this conclusion). We haven't changed the text, however, as we feel a stronger statement would be beyond the scope of this manuscript.

TABLES AND FIGURES:

Table 2: I would be interested to these **results for N3 and N4** (according to pre-2008 scoring standards) separately, if there is any such data available, since there are real differences between the stages.

See above: we've now included a new analysis of spindles during N4 sleep.

In general, I think it would be helpful to consistently **provide units** with all of the measures in all the figures, despite many of them being the same throughout.

We've added units to figure/table legends in places where it was ambiguous.

Figure 3: It seems like the lowest bin for the spindle duration histogram is for durations between 0.5 and 0.6 s, so it seems **somewhat confusing start the x-axis at 0.6**. Also, this figure is nice, but I would like to see some of this general information on the properties of spindles across the population tabulated too.

Agreed, we've amended the labeling for this Figure. This information is tabulated in Table S2.

Figure 4: I would move the x-axis labels to below the female plots.

Agreed: labels are now duplicated beneath both male and female plots. (Note: this Figure is now Supplementary, Figure S10).

Figure 7: It's stated in the caption, but I think a legend for the color of the lines corresponding to the two racial groups would also be helpful.

Done. (Note: this has been moved to the Supplement, and is now Figure S22).

Figure 8: I would move the sentence "Note, because of the normalization, the profile..." to before the sentence "The third column shows,..." in this caption. Currently it seems to be referring to the plots in the third column, although I don't think that is the case.

This Figure has been removed. (Figure S21 contains the same information; the text about normalization for that legend has been changed as suggested.)

Figure 9: The asterisk on the non-significant p-value for the SOF comparison is not actually defined.

Corrected.

Figure 10: There should be a color scale for the r_g plots. There's also an extra hyphen at the end of the second line of the caption.

Agreed, corrected and legend added.

Reviewer #3 (Remarks to the Author):

This is weighty article in which a huge effort has been made for describing, for providing normative values, and for defining "the state of art" of our knowledge on sleep spindles (SS). The other face of this article is that majority of findings are not original.

The evaluation of a sleep researcher qualifies it as a probable milestone in studies on SS. On the whole, I have found it intellectually honest.

As known by the authors, the principle limit of the study is represented by the **lack of any regional (topographical) information**. In fact, most of well decribed phenomena of the current study interact or covary with regional changes. This is not merely a missing finding.

See above. We have addressed this issue by adding a new section that considers the 53 CHAT individuals who have fuller (18 channel) EEG recordings. We focus on the relationship between individual differences in spindle activity observed at C3/C4 versus the other sites, to inform what would be captured or missed by an analysis of only C3/C4.

Another important missing issue regards **the circadian modulation of SS** (e.g., Knoblauch et al. Sleep. 2005; 28:1093-101; Knoblauch et al., Eur J Neurosci. 2003;18:155-63; Wei et al., Neurosci Lett. 1999; 260: 29-32). At variance of topograhly, I suppose that the authors can make an effort in this direction, and they have to select data collected in different times of day/night.

Agreed. See above: as well as revising the analysis of spindle dynamics to focus on sleep cycles and other ultradian factors (NREM/REM transitions, etc), we have included an analysis of putative circadian modulation as well as chronotype (indexed by sleep mid-point).

The results are intriguing and add an interesting new element to the manuscript; as noted above, we replicate the previous observation of distinct circadian modulation of fast versus slow spindles, and of the attenuation of circadian modulation as a function of increasing age. We comment on how this approach could be extended to score individuals for their extent of circadian modulation, which could be used as a phenotype in genetic studies, for example. Genetic influences on the circadian modulation of spindles could conceivably be distinct from genetic influences on baseline spindle activity, or on other ultradian aspects of spindle dynamics.

Moving to the specific points:

1. In my opinion, the analyses on the spindle density and sleep stage duration could be drastically reduced due to the fact that they are mostly obvious;

We've shortened the manuscript in a number of places, removing unnecessary Tables and Figures. We feel it is important to report briefly on spindles and sleep marco-architecture, however, as these are key properties of the sample and the pipeline, even if some points are obvious to some readers.

2. Data on inter-hemispheric EEG (C3, C4) spectral coherence strictly resemble those on raw EEG power values (Figure S11). This deserves a comment.

This Figure has been removed from the manuscript. (We assume the implied comment would be to suggest that the increased coherence reflects spindle activity? If so, this alluded to in Table S27.)

3. The lack of evidence for a bimodal distribution of spindle frequency seems misleading. It seems a by-product of: (A) the lack of frontal recordings (i.e., the "lacking" second peak is maximal at these leads), (B) the fact that the "canonical" analysis is centered on 13.5 Hz;

We feel that showing the functional dissociations between fast and slow spindles (i.e. in terms of dynamics, topography, etc) is a compelling demonstration that fast and slow spindles are different (as others have shown). As we note, there is necessarily some "bleed through" of spindles outside of the targeted frequency range. Although various factors are clearly associated with spindle frequency, we agree with Reviewer 1 that a precise delineation (*exactly two classes? or a more continuous range? etc*) is difficult and beyond the scope of the current manuscript.

4. The large difference between "frequency-independent" and "frequency-dependent" estimates (7.4 vs. 1.88) should be better clarified

We've extended the previous text that comments on this, that now reads: "Because some slower spindles aren't detected with higher values of F_C , and vice versa, this 'total' spindle density estimate is naturally higher than the canonical estimate, which only targeted the smaller, core range of spindle frequencies."

5. When describing life course trajectories of frequency-dependent spindle density, the authors conclude that "it will be challenging to interpret changes in spindle activity if age and spindle frequency are not taken into account". This is true, but even more is challenging if age and spindle frequency do not **take into account regional changes** (e.g. for infants: Novelli et al., J Sleep Res. 2016; 25: 381-9; for elderly: Martin et al., Neurobiol Aging. 2013; 34: 468-76). Please, discuss in some way also this issue

We largely agree and have added explicit text to address this, as well as the new topographical analyses (i.e. Figure 6, S24, S25). We do show that – in children at least – the lion's share of inter-individual within-channel variation in spindle activity is captured by fast and slow spindles at central electrodes. At the same time, we also now more explicitly note the limitations of spindle studies that lack fuller topographical information.

6. Analyses regarding spindle trajectory over the course of the night are suboptimal. Please, **plot changes of spindles across consecutive sleep cycles/periods**. Sleep periods more than time spent in N2 sleep describes the time course of SS across the night. Furthermore, this approach will make results more comparable to those previously published

Agreed. See above: we have now reanalyzed the data in terms of sleep cycles. We agree this is a preferable manner for presenting the results. However, consistent with our original observations, we go on to show that other ultradian factors, including within-cycle variation, as well as elapsed sleep are nonetheless significant factors in spindle dynamics (i.e. S18), over and above sleep cycle number.

7. Please, de-emphasize relevance of genetic results. They do not add so much to the existing knowledge (e.g., Ambrosius et al., Biol Psychiatry. 2008; 64: 344-8; De Gennaro et al., Neuroimage. 2005; 26: 114-22; De Gennaro et al., Ann Neurol. 2008; 64: 455-60; Adamczyk et al., Front Hum Neurosci. 2011; 9: 624)

We've changed the text to more clearly frame our contribution in the light of the previous literature. (Note that all the above references were cited in the original manuscript.)

Again, we thank all reviewers for their helpful comments.

REVIEWERS' COMMENTS:

Reviewer #1 (Remarks to the Author):

The readability of this new version of the manuscript has greatly improved. Although it is still quite long, it is now a very nice and informative read.

It would be preferable to use S2-S4 instead of SWS2-SWS4 for the R&K analysis.

Although it should be clear, mention "age" in Figure S18b or use labels as in Fig. S19.

Consistency between Tables showing regression weights b and p values could still be improved. Table S27 should indicate predictors. Table S11 might be structured like Tables S22 or S27.

In Figure S25, right top and bottom halves should be switched to match frequency labels.

The authors should again confirm that their average spindle density is 1.88 per minute and not per 30-s epoch. From my experience, stable stage 2 sleep in young adults usually shows more than 1 clearly visible spindle per 30-s epoch.

How does Table S30 relate to the mean spindle density of 1.88 of the canonical 13.5-Hz analysis or those reported for CHAT, SHHS and MrOS?

Table S32 could provide b as well as p values. Also, negative p values can be confusing.

In Figure S27, "T1" threshold is probably supposed to mean "TL".

My personal opinion is that after showing lower test/retest reliability in Table S30, further analyses with lower detection thresholds are unnecessary (Tables S31, S32, S33 and Figures S27, S28) and could be removed. I am not opposed to keeping these items, but in my opinion these analyses do not provide clear evidence and are therefore of less value than the rest of the paper.

Regarding Figure S28, I still do not see how physiological spindles would be generated in a way that results in the bottom panel (linear shift to right). This distribution would require a large-or-nothing response that is always above a certain voltage threshold when triggered. I am not aware of such a mechanism in sleep spindle generation. Correspondingly, I have never seen a lower cutoff amplitude for sleep spindles, but in my experience there are always spindles of all amplitudes present, from "barely visible", "just below/above threshold" to "clearly exceeding threshold". And even if physiology would allow a lower amplitude threshold for sleep spindles, detection mechanisms would not. Any noisy $1/f$ signal like the EEG will always show a certain number of events of a particular amplitude. Thus, I would expect a (left-skewed) broader distribution with a peak at 45 μV and a maximum of 60 μV , but reaching down to 0 μV .

Reviewer #2 (Remarks to the Author):

The authors have adequately addressed all of my previous concerns and moreover have greatly improved the paper. The reduced length is beneficial, but the added analyses are also valuable; I am particularly glad to see the added analysis of spindles occurring during N4 sleep.

One issue: the caption for Figure 6 a) and b) doesn't quite seem to match the figure—the colormap is black/red/yellow/white, not blue/white/red, and I don't see any diamonds, only circles which I interpret as representing only the electrode locations.

Other small typos/errors I noticed while reading through:

Page 9, paragraph 2, line 5: "particular" should be "particularly"

Page 9, paragraph 3, line 6: "spindles density" should be "spindle density"

Page 10, paragraph 4, line 1: "to" should be removed

Page 22, paragraph 4, line 6: "reduce" should be "reduced"

Reviewer #3 (Remarks to the Author):

In my opinion, all points I raised in the previous round of review have been satisfactorily addressed. The authors have done an excellent work, improving/refining their colossal study.

I particularly appreciated their effort to respond to the questions regarding (A) the dynamics of sleep spindles across sleep cycles; (B) the evaluation of the circadian effects [although the intrinsic limit of not having data coming from appropriate experimental protocols (i.e., forced desynchrony or constant routine)]; (C) some specific topographical aspects.

Summarizing, my opinion is favourable, and I think that it will become a probable milestone in studies on sleep spindles, also giving new research ideas.

Luigi De Gennaro

(I have no problem or need of anonymity regarding) my previous detailed evaluation)

Response to Reviews of the Revised Manuscript

We again thank all three reviewers. Below, reviewers' comments are in *gray italics*, our responses in this font.

Reviewer #1

The readability of this new version of the manuscript has greatly improved. Although it is still quite long, it is now a very nice and informative read.

It would be preferable to use S2-S4 instead of SWS2-SWS4 for the R&K analysis.

We use labels N1, N2 and N3 to refer to sleep stages, with the exception of one analysis that focuses on the older staging that includes NREM4, following the standard nomenclature. We are not sure what this comment refers to, as we don't use the abbreviation "SWS" in the manuscript.

Although it should be clear, mention "age" in Figure S18b or use labels as in Fig. S19.

Agreed, fixed in the Figure.

Consistency between Tables showing regression weights b and p values could still be improved. Table S27 should indicate predictors. Table S11 might be structured like Tables S22 or S27.

Agreed, we've restructured Table S11.

In Figure S25, right top and bottom halves should be switched to match frequency labels.

Agreed, thank you for spotting this error in assembling the Figure.

The authors should again confirm that their average spindle density is 1.88 per minute and not per 30-s epoch. From my experience, stable stage 2 sleep in young adults usually shows more than 1 clearly visible spindle per 30-s epoch.

Confirmed: however, we underscore that the estimated rate is naturally method-dependent and threshold-dependent (which motivated the subsequent analyses varying thresholds).

How does Table S30 relate to the mean spindle density of 1.88 of the canonical 13.5-Hz analysis or those reported for CHAT, SHHS and MrOS?

This is a good observation, that indeed deserves explanation. We've added text to the legend to flag and clarify this difference, which we agree may otherwise seem confusing. The bottom line is that the apparent differences reflect the parameter-dependent nature of any spindle detection algorithm.

As the Reviewer notes, with the default threshold ($t = 4.5$) we report a density of 2.7 in Table S30, for $F_C=13\text{Hz}$. In contrast, the "canonical" analysis reports a density of 1.88. The difference is driven by the bandwidth (i.e. number of cycles for the wavelet) used in the frequency-dependent analyses versus the canonical analysis (12 versus 7, as we noted in the Methods section). We increased the number of cycles in the frequency-dependent analyses to provide greater frequency resolution, at the expense of temporal resolution. (Note: the difference between F_C of 13 Hz versus 13.5 Hz does not have a great impact here.) The correlation between these two estimates, which is arguably the more relevant comparison for studies of individual differences, is still very high however, at $r \sim 0.9$.

On face value, it does appear counter-intuitive that the “broader” canonical analysis captures fewer spindles than one more narrowly focused around the target frequency. However, it is important to note that wavelet parameters will impact the baseline distribution of wavelet coefficients, which includes sample points “under the null” (i.e. when there is no targeted spindle present), as well as the when targeted spindles are actually present. As such, both “signal” and “noise” components will be influenced by parameter selection: the resulting impact on sensitivity and specificity may not be straightforwardly predictable.

In the absence of any clear gold standard, these mean differences are not of particular concern. Rather, they simply underscore the parameter-dependent nature of spindle detection. Indeed, as the Table already shows, a different amplitude threshold ($t=1$) leads to a still higher estimate of density (>8 spindles per minute) for ~13 Hz spindles.

Table S32 could provide b as well as p values. Also, negative p values can be confusing.

Agreed, we’ve added regression coefficients to explicitly represent the direction of effect and use standard instead of log notation for the p -values.

In Figure S27, “T1” threshold is probably supposed to mean “TL”.

Agreed, fixed.

My personal opinion is that after showing lower test/retest reliability in Table S30, further analyses with lower detection thresholds are unnecessary (Tables S31, S32, S33 and Figures S27, S28) and could be removed. I am not opposed to keeping these items, but in my opinion these analyses do not provide clear evidence and are therefore of less value than the rest of the paper.

We agree that, as reported, the signal-to-noise ratio is lower for “low-amplitude” spindles. Nonetheless, that we observe very highly significant (i.e. $p = 10^{-117}$) associations with demographic factors such as age suggests there is still sufficient signal to justify further analyses. Furthermore, that low-amplitude spindles show highly significant associations but qualitatively different directions of effect compared to the default analyses deserves explanation, thus the additional analyses. The focus on low-amplitude spindles also makes some broader points with respect to the parameter-dependent nature of spindle detection, and the inter-dependency of spindle metrics, which we still think constitute some useful take-home messages.

Regarding Figure S28, I still do not see how physiological spindles would be generated in a way that results in the bottom panel (linear shift to right). This distribution would require a large-or-nothing response that is always above a certain voltage threshold when triggered. I am not aware of such a mechanism in sleep spindle generation. Correspondingly, I have never seen a lower cutoff amplitude for sleep spindles, but in my experience there are always spindles of all amplitudes present, from “barely visible”, “just below/above threshold” to “clearly exceeding threshold”. And even if physiology would allow a lower amplitude threshold for sleep spindles, detection mechanisms would not. Any noisy 1/f signal like the EEG will always show a certain number of events of a particular amplitude. Thus, I would expect a (left-skewed) broader distribution with a peak at 45 μ V and a maximum of 60 μ V, but reaching down to 0 μ V.

Agreed. (We perhaps misunderstood the original comment made with respect to this Figure. In fact, the reviewer’s comments echo our original concerns with some standard practices in spindle detection, whereby stringent detection thresholds effectively assume/imply a discontinuous, all-or-nothing distribution of spindle amplitudes.)

The motivation behind the Figure was to show that density and amplitude measurements may have counter-intuitive relationships, depending on detection thresholds. We appreciate that the concrete depiction of an underlying amplitude distribution (which, as we stated in the legend is naturally

unknown) can cause confusion, however. For instance, it implicitly makes a statement about the proportion of “true spindles” that are detected, which we of course do not know. Our focus was on the interpretation of the qualitative differences between the three panels. Nonetheless, we agree with the reviewer’s concerns, and have replaced the Gaussian distributions with non-central chi-squared distributions that include 0 in all cases, and so should not be taken to imply an “all-or-nothing” phenomenon.

Reviewer #2

The authors have adequately addressed all of my previous concerns and moreover have greatly improved the paper. The reduced length is beneficial, but the added analyses are also valuable; I am particularly glad to see the added analysis of spindles occurring during N4 sleep.

Thank you.

One issue: the caption for Figure 6 a) and b) doesn’t quite seem to match the figure—the colormap is black/red/yellow/white, not blue/white/red, and I don’t see any diamonds, only circles which I interpret as representing only the electrode locations.

Corrected.

Other small typos/errors I noticed while reading through:

Page 9, paragraph 2, line 5: “particular” should be “particularly”

Corrected.

Page 9, paragraph 3, line 6: “spindles density” should be “spindle density”

Corrected.

Page 10, paragraph 4, line 1: “to” should be removed

Corrected.

Page 22, paragraph 4, line 6: “reduce” should be “reduced”

Corrected.

Reviewer #3

In my opinion, all points I raised in the previous round of review have been satisfactorily addressed. The authors have done an excellent work, improving/refining their colossal study. I particularly appreciated their effort to respond to the questions regarding (A) the dynamics of sleep spindles across sleep cycles; (B) the evaluation of the circadian effects [although the intrinsic limit of not having data coming from appropriate experimental protocols (i.e., forced desynchrony or constant routine)]; (C) some specific topographical aspects.

Summarizing, my opinion is favourable, and I think that it will become a probable milestone in studies on sleep spindles, also giving new research ideas.

Luigi De Gennaro

(I have no problem or need of anonymity regarding) my previous detailed evaluation)

Thank you, no response needed.